



# Correcting model biases of CO in East Asia: impact on oxidant distributions during KORUS-AQ

Benjamin Gaubert[1], Louisa K. Emmons[1], Kevin Raeder[2], Simone Tilmes[1], Kazuyuki Miyazaki[3], Avelino. F. Arellano Jr[4], Nellie Elguindi[5], Claire Granier[5,6], Wenfu Tang[7], Jérôme Barré[8], Helen M. Worden[1], Rebecca R. Buchholz[1], David P. Edwards[1], Philipp Franke[9], Jeffrey L. Anderson[2], Marielle Saunois[10], Jason Schroeder[11], Jung-Hun Woo[12]. Isobel J. Simpson[13], Donald R. Blake[13], Simone Meinardi[13], Paul O. Wennberg[14], John Crounse[14], Alex Teng[14], Michelle Kim[14], Russell R. Dickerson[15,16], Hao He[15,16], Xinrong Ren[15,17]

[1]Atmospheric Chemistry Observations and Modeling, National Center for Atmospheric Research, Boulder, CO, USA.
[2]Computational and Information Systems Laboratory, National Center for Atmospheric Research, Boulder, CO, USA.
[3]Jet Propulsion Laboratory, California Institute of Technology, Pasadena, CA, USA.
[4]Dept. of Hydrology and Atmospheric Sciences, University of Arizona, Tucson, AZ, USA.
[5]Laboratoire d'Aérologie, CNRS, Université de Toulouse, France.
[6]NOAA Chemical Sciences Laboratory-CIRES/University of Colorado, Boulder, CO, USA.
[7]Advanced Study Program, National Center for Atmospheric Research, Boulder, CO, USA.
[8]European Centre for Medium-Range Weather Forecasts, Shinfield Park, Reading, RG2 9AX, UK.
[9]Forschungszentrum Jülich GmbH, Institut für Energie und Klimaforschung IEK-8, 52425 Jülich, Germany.
[10]Laboratoire des Sciences du Climat et de l'Environnement, LSCE-IPSL (CEA-CNRS-UVSQ), Université Paris-Saclay, 91191 Gif-sur-Yvette, France.
[11]California Air Resources Board, Sacramento, CA, USA.
[12]Department of Advanced Technology Fusion, Konkuk University, Seoul, South Korea.
[13]Department Chemistry, University of California, Irvine, Irvine, CA 92697, USA.
[14]California Institute of Technology, Pasadena, CA, USA.
[15]Department of Atmospheric and Oceanic Science, University of Maryland, College Park, MD, USA
[16]Earth System Science Interdisciplinary Center, University of Maryland, College Park, MD, USA
[17]Air Resources Laboratory, National Oceanic and Atmospheric Administration, College Park, MD, USA

*Correspondence to*: Benjamin Gaubert (gaubert@ucar.edu)

**Abstract.** Global coupled chemistry-climate models underestimate carbon monoxide (CO) in the Northern Hemisphere, exhibiting a pervasive, negative bias against measurements peaking in late winter and early spring. While this bias has been commonly attributed to underestimation of direct anthropogenic and biomass burning emissions, chemical production and loss via OH reaction from emissions of anthropogenic and biogenic VOCs play an important role. Here we investigate the reasons for this underestimation using aircraft measurements taken in May and June 2016 from the Korea United States Air Quality (KORUS-AQ) experiment in South Korea and the Air chemistry Research In Asia (ARIAs) in the North China Plain (NCP). For reference, multispectral CO retrievals (V8J) from the Measurements of Pollution in the Troposphere (MOPITT) are jointly assimilated with meteorological observations using an Ensemble Adjustment Kalman Filter (EAKF) within the global Community Atmosphere Model with Chemistry (CAM-chem) and the Data Assimilation Research Testbed (DART). With regard to KORUS-AQ data, CO is underestimated by 42 % in the Control-Run and by 12 % with the MOPITT assimilation run. The





inversion suggests an underestimation of anthropogenic CO sources in many regions, by up to 80
% for Northern China, with large increments over the Liaoning province and the North China
Plains (NCP). Yet, an often-overlooked aspect of these inversions is that correcting the
underestimation in anthropogenic CO emissions also improves the comparison with observational
$O_3$ datasets, and observationally constrained box model simulations of OH and $HO_2$. Running a
CAM-chem simulation with the updated emissions of anthropogenic CO reduces the bias by 29 %
for CO, 18 % for ozone, 11 % for $HO_2$ and 27 % for OH. Longer lived anthropogenic VOCs whose
model errors are correlated with CO are also improved while short-lived VOCs, including
formaldehyde, are difficult to constrain solely by assimilating satellite retrievals of CO. During an
anticyclonic episode, better simulation of $O_3$, with an average underestimation of 5.5 ppbv and a
reduction in the bias of surface formaldehyde and oxygenated VOCs can be achieved by separately
increasing by a factor of two the modeled biogenic emissions for the plant functional types found
in Korea. Results also suggest that controlling VOC and CO emissions, in addition to wide spread
NOx controls, can improve pollution ozone over East Asia.

## 1 Introduction

Carbon monoxide (CO) is a good tracer of biomass burning (Edwards et al., 2004; Edwards et al.,
2006) and anthropogenic emission sources (e.g. Borsdorff et al., 2019). It is also the main sink of
the hydroxyl radical (OH) and therefore is important in quantifying the methane ($CH_4$) sink in the
troposphere (Myhre et al., 2013; Gaubert et al., 2016, 2017; Nguyen et al., 2020). In fact, because
of the lack of observational constraints on the OH spatio-temporal variability, uncertainties in the
atmospheric $CH_4$ lifetime and its interannual variability have precluded accurately closing the
global $CH_4$ budget (Saunois et al., 2016; Prather & Holmes, 2017; Turner et al., 2019). There is a
need to reduce uncertainties in the main drivers of OH (National Academies of Sciences,
Engineering, and Medicine 2016), which are CO, ozone ($O_3$), water vapor ($H_2O$), nitrogen oxides
($NO_x$), and non-methane volatile organic compounds (NMVOCs).

The evolution of CO in Eulerian chemical-transport is governed for each grid cell by Eq. (1):

$$\frac{\delta CO}{\delta t} = -v \cdot \nabla[CO] + \sum_{i=1}^{Sectors} E_i + \sum_{i=1}^{Chemicals} \chi_i - k[CO][OH] - k_{deposition}[CO] \tag{1}$$

CO has only one chemical sink, its reaction with OH ($k[CO][OH]$). The other CO sink is dry
deposition ($k_{deposition}[CO]$) through soil uptake (Conrad, 1996; Yonemura et al., 2000; Stein et
al., 2014, Liu et al., 2018). The direct sources are the emissions from different sectors $E_i$, the
anthropogenic (fossil fuel and biofuel), biomass burning, biogenic and oceanic sources. Locally,
CO can be advected from neighboring grid cells ($-v \cdot \nabla[CO]$) and produced from the oxidation of
NMVOCs ($\chi_i$). Globally, the oxidation of $CH_4$ is the main source of chemically produced CO.
Biogenic and anthropogenic NMVOCs also contribute significantly to secondary CO.

The use of inverse models and chemical data assimilation systems has helped in constraining the
global CO budget and associated trends at global to continental scales, particularly with the
availability of long time series of CO retrievals from the Measurement of Pollution In the
Troposphere (MOPITT, Worden et al., 2013) satellite instrument (e.g., Arellano et al., 2004;
Pétron et al., 2004; Heald et al. 2004; Kopacz et al., 2010; Fortems-Cheiney et al., 2011). Such
studies are generally in agreement with regards to the decreasing long-term trends in CO emissions
from anthropogenic and biomass burning sources (Jiang et al. 2015; Yin et al., 2015; Miyazaki et
al. 2017; Zheng et al., 2019), although regional emissions remain largely uncertain. Outstanding



issues reported in the literature that still need to be resolved include errors in model transport (Arellano and Hess 2006; Jiang et al. 2013), lack of accurate representation of the atmospheric vertical structure of CO (Jiang et al., 2015) and OH fields (Jiang et al., 2011; Müller et al., 2018),

aggregation errors (Stavrakou and Müller, 2006; Kopacz et al., 2009), and inclusion of chemical feedbacks (Gaubert et al., 2016). Recent studies have suggested mitigating these issues by assimilating multiple datasets of chemical observations (Pison et al. 2009; Fortems-Cheiney et al. 2012; Kopacz et al., 2010; Miyazaki et al., 2012; Miyazaki et al., 2015), and the use of different models that use the same data assimilation system (Miyazaki et al., 2020a).


Regionally, it has become evident from both forward and inverse modeling approaches that several standard inventories of CO emissions in China are still too low (e.g. Tang et al., 2013; Yumimoto et al. 2014). Recently, Kong et al. (2020) compared a suite of 13 regional model simulations with surface observations over the North China Plain (NCP) and Pearl River Delta (PRD) and found a

severe underestimation of CO, despite the models using the most up-to-date emissions inventory, the mosaic Asian anthropogenic emission inventory (MIX) (Li et al., 2017). Using surface CO observations in China, Feng et al. (2020) performed an inversion of the MIX inventory and found posterior emissions that were much higher than the priors, with regional differences, still pointing to a large underestimation in northern China. The large posterior increase of CO emissions in

northern China seems to be due to a severe underestimation of residential coal combustion for heating and potentially for cooking (Chen et al., 2017; Cheng M., et al., 2017; Zhi et al., 2017).

While the general underestimation of fossil fuel burning in East Asia seems to explain the underestimation of Northern Hemisphere (NH) extratropical CO found in global models (Shindell et

al., 2006), there are other confounding factors. Naik et al. (2013) found large inter-model variability in the regional distribution of OH and an overestimation of OH in the NH. This is consistent with an overestimation of ozone (Young et al. 2013), which provides another explanation of the CO underestimation. Strode et al. (2015) confirmed that the springtime low bias in CO is likely due to a bias in OH. This can be caused by a bias in ozone and water vapor, which

are OH precursors. Yan et al. (2014) suggested that these biases could be mitigated by increasing the horizontal resolution within a 2-way nested model. Stein et al. (2014) suggested that anthropogenic CO and NMVOCs from road traffic emissions were too low in their inventory, but also suggested that a wintertime increase in CO could be due to a reduced deposition flux. Secondary CO originating from the oxidation of $CH_4$ and NMVOCs could also play a role in the

CO underestimation (e.g. Gaubert et al., 2016).

Due to significant efforts in reducing emissions in China, including effective implementation of clean air policies which started in 2010 (e.g. Zheng et al., 2018), there has been a reduction of CO emissions of around 27 % since 2010. Bhardwaj et al. (2019) found a decrease of surface MOPITT

CO by around 10 % over the North China Plain and South Korea during the 2007-2016 period. While NOx emissions have been decreasing since 2010, the net NMVOCs emissions have been increasing (Zheng et al., 2018), and satellite observations show a positive trend in $CH_2O$ (Shen et al., 2019). Li, M. et al. (2019) found an increase in NMVOCs emissions from the industry sector and solvent use while emissions from the residential and transportation sectors declined, leading

to a net increase in emissions of NMVOCs. A modeling study suggests that the reduction of aerosols over northern China has reduced the sink of hydroperoxyl radicals ($HO_2$) which resulted in an increase in surface $O_3$ concentrations in North Eastern China (Li, K. et al., 2019). Ozone production and accumulation of precursors in China is often associated with strong sunlight and low winds, similar to other parts of the world, but wind direction is also particularly important

because it affects pollution transport (Wang et al., 2017).



Emissions from East Asia are known to impact regional air quality (AQ), and contribute significantly to surface $O_3$ pollution at regional, continental and even intercontinental scales through trans-Pacific transport, in particular in spring when meteorological conditions favor rapid transport (Akimoto et al., 1996; Jacob et al., 1999; Wilkening et al., 2000; Heald et al., 2006). Frontal lifting in warm conveyor belts (WCBs) efficiently contributes to the transport of pollution (Cooper et al. 2004; Zhang et al. 2008; Lin et al. 2012), which can be observed by satellite retrievals of tropospheric $O_3$ (Foret et al., 2014) and aircraft in-situ measurements (Ding et al. 2015). However, the mechanisms that cause the uplifted pollution to effectively descend to the downwind surface layers at regional, continental and intercontinental scales are complex. In the case of South Korea, one efficient mechanism could be that once lifted from the emission sources in China, the higher altitude plumes can pass through the marine atmosphere of the Yellow Sea without removal processes such as dry deposition, and reach the surface of the Korean peninsula during the day, when the boundary layer is high (Lee et al., 2019a; Lee et al., 2019b). In addition, severe pollution episodes can be due to local emissions under stagnant conditions with reduced regional ventilation and lower wind speed (Kim et al. 2017).

The recent literature and findings from the 2016 field campaign over South Korea indicate the relative importance of $O_3$ precursors and associated transport in this region. The Korea-United States Air Quality (KORUS-AQ) field campaign was a joint effort between the National Aeronautics and Space Administration (NASA) of the United States and the National Institute of Environmental Research (NIER) of South Korea. The field campaign's objective was to quantify the drivers of AQ over the Korean Peninsula with a focus on the Seoul Metropolitan Area (SMA), currently one of the largest cities in the world. The intensive measurement period was from May 1 2016 and June 15 2016 with the deployment of a research vessel (Thompson et al., 2019) and 4 different aircraft: the NASA DC-8, the NASA B200, the Hanseo University King Air and the Korean Meteorological Agency (KMA) King Air. The aircraft sampled numerous vertical profiles of trace gases, aerosols and atmospheric physical parameters with missed approach flying procedure over the SMA (e.g. Nault et al., 2018) and spiral patterns over the Taehwa Research Forest (TRF) site, downwind from the SMA (e.g. Sullivan et al., 2019). Peterson et al. (2019) studied the weather patterns during KORUS-AQ and distinguished four distinct periods defined by different synoptic patterns: a dynamic meteorological phase with complex aerosol vertical profiles, a stagnation phase with weaker winds, a phase of efficient long-range transport, and a blocking pattern.

This campaign provides several case studies of foreign-influenced and local pollution episodes. Miyazaki et al. (2019a) assimilated a suite of satellite remote sensing of chemical observations and found that under dynamic conditions, when there was efficient transport with uplifting of pollution to higher altitudes (where the satellite has more sensitivity), forecasted ozone was improved by the assimilation of satellite ozone retrievals. On the contrary, under stagnant conditions, forecasted ozone was not improved as much when compared to the DC-8 ozone measurements, suggesting ozone formation closer to the surface. Lamb et al. (2018) looked at the vertical distribution of black carbon during KORUS-AQ. Aside from a short episode of biomass burning sources from Siberia, they found that the Korean emissions were important in the boundary layer, with a large contribution from long-range transport from mainland China that varies with the large-scale weather patterns. There are different ways to quantify the sources contributing to pollutants, such as Lagrangian back trajectory, VOCs signatures, CO to $CO_2$ ratios and CO tags (Tang et al., 2019). Overall, direct Korean CO emissions are important contributors to the boundary layer CO, but not higher up where emissions from continental Asia dominate. This suggests an underestimation of





emissions, including those from the chemical oxidation of VOCs. Simpson et al. (2020) performed
a source apportionment of the VOCs over the SMA and also found a significant source of CO from
long-range transport with only a smaller CO source from combustion over Seoul. Since long-range
transport is important, the forecasted CO and water vapor during KORUS-AQ can be improved
by assimilating Soil Moisture from the NASA SMAP satellite (Soil Moisture Active Passive) over
China (Huang et al, 2018). They stress the importance of error sources stemming from chemical
initial and boundary conditions and emissions for modeling CO during two episodes of the
transport phase.

While chemical data assimilation is efficient for CO in a global model, because of its longer
lifetime than most of the reactive species, there are some limitations if the parameters, such as
emissions inventories inputs or physical and chemical processes, are not updated consistently with
the initial conditions. The KORUS-AQ campaign provides a large array of measurements and is
an excellent case study for testing the model with challenges that need to be addressed for further
improvements of CO and related species of interest such as OH, $O_3$, $CH_4$ and NMVOCs. Here we
take advantage of the concurrent measurements during the campaign to investigate the reasons for
the CO underestimation and we attempt to answer the following question: Can we explain why
CO is consistently underestimated over East Asia, using a Chemical Transport Model, field
campaigns and satellite data assimilation?

We outline the set of observations used to verify and evaluate our chemical data assimilation
system in Section 2. The modeling system is presented in Section 3, the Data Assimilation system
in Section 4, the evaluation of the data assimilation results in Section 5. The comparison of
emissions estimates and additional sensitivity experiments in Section 6.

**2 Field campaign observations**

**2.1 The Korea United States Air Quality (KORUS-AQ) field campaign**

The KORUS-AQ campaign provides a unique testbed for comparing surface and aircraft in-situ
observations with ground-based and satellite-based remote sensing (Herman et al. 2018),
particularly important for the targeted short-lived species such as formaldehyde ($CH_2O$) and
nitrogen dioxide ($NO_2$). Miyazaki et al. (2019a) showed that the background $O_3$ measured by the
DC-8 during KORUS-AQ ranges from 72 to 85 ppbv between the surface and 800 hPa over the
Korean Peninsula. On top of these large background values, large emissions from the SMA are
responsible for the strong formation of secondary organic aerosols (Kim et al., 2018; Nault et al.,
2018) and $O_3$, which can be further enhanced by biogenic emissions eastward of Seoul (Sullivan
et al., 2019). A high correlation between organic aerosol and $CH_2O$, which is a characteristic of
the importance of primary and secondary sources, was found during the campaign (Liao et al.,
2019). Large ozone production is a result of emissions from areas characterized to be VOC-limited,
such as the urbanized SMA and industrialized regions into a $NO_x$-limited environment over rural
and forests area. Both Oak et al. (2020) and Schroeder et al. (2020) examined $O_3$ production during
KORUS-AQ with a focus on the SMA and surrounding regions and reported a higher ozone
production efficiency over the rural areas. They pointed out higher ozone sensitivity to aromatics,
followed by isoprene and alkenes. Observations over the Taehwa Research Forest east of Seoul
show strong ozone production (Kim et al., 2013) because of large emissions of reactive biogenic
VOCs, in particular isoprene and monoterpenes.




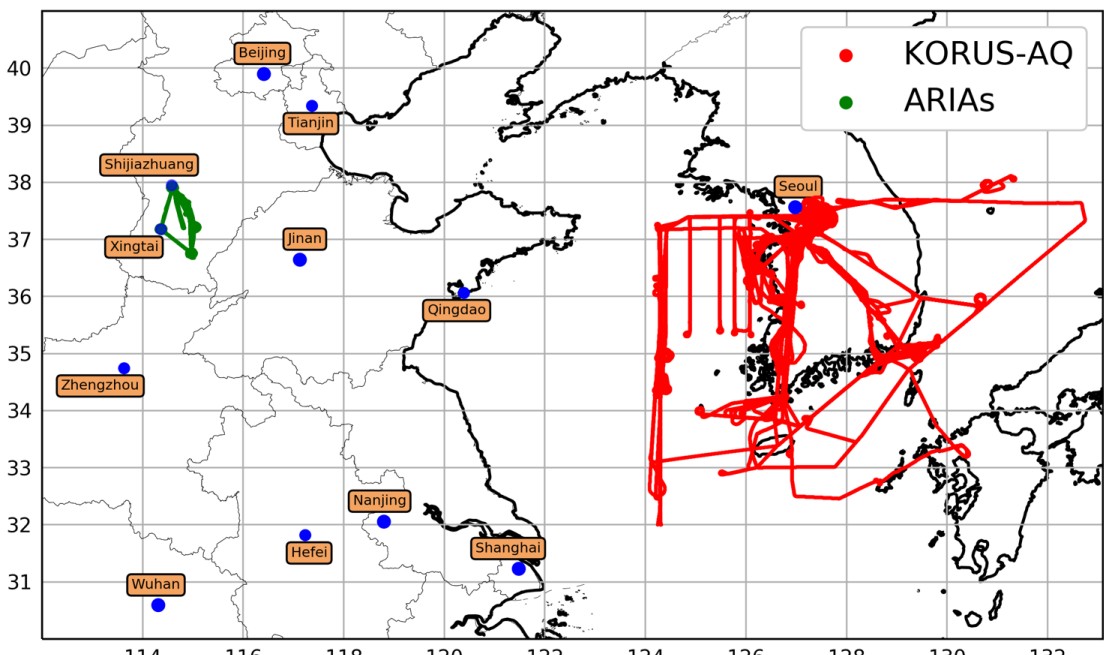

**Figure 1: Location of all the KORUS-AQ DC-8 1-min merge measurements (red dots), and of the ARIAs Y-12 1-min merge measurements (green dots). The location of some major cities is also indicated (blue dots).**

We evaluate the model output against the DC-8 aircraft measurements, shown in red in Figure 1, which simultaneously provide many physical and chemical parameters of the tropospheric chemistry environment system (appendix A). we use the 1-minute merge file of DC-8 in-situ observations. Model outputs were linearly interpolated to the exact location of the DC-8 in latitude, longitude, pressure altitude and in time, from the 6 hourly model outputs. During the whole

campaign, Simpson et al. (2020) showed that high benzene concentrations (> 1 ppbv) were only found close to the Daesan petrochemical complex. Since those large gradients of local plumes simply cannot be modeled in a global model, we systematically rejected observations when the benzene proton-transfer-reaction time-of-flight mass spectrometer (PTR-ToF-MS) measurements were higher than 1 ppb.


In order to evaluate the CO sink and the impact of the assimilation of MOPITT CO retrievals on the $HO_x$ levels, we used the OH and $HO_2$ calculated with the NASA Langley Research Center (LaRC) 0-D time-dependent photochemical box model (Schroeder et al., 2020). This box model is constrained by measured temperature and pressure, photolysis rates derived from actinic flux

observation and observations of $O_3$, NO, CO, $CH_4$, $CH_2O$, PAN, $H_2O_2$, water vapor, and non-methane hydrocarbons. The production and loss terms of ozone is calculated for every single 1 Hz DC-8 set of observations. This is the only case where we use the 1-second merge file instead of the 1-min merge dataset file. CAM-chem outputs are interpolated accordingly. While there are some limitations for the species with longer lifetimes, subject to physical processes that are not

represented, the box model has been specifically designed to estimate radical concentrations. The details and sensitivity of the calculation are described in Schroeder et al. (2020).





## 2.2 The ARIAs campaign

The Air Chemistry Research In Asia (ARIAs) field campaign was conducted in May and June 2016 with the goal of better quantifying and characterizing air quality over the North China Plain (Benish et al., 2020; Wang et al. 2018). The instrumented Y-12 airplane was operated by the Weather Modification Office of the Hebei Province to measure meteorological parameters, aerosols optical properties and trace gases. The airplane was based in Luancheng Airport, southeast of Shijiazhuang, the capital of Hebei Province, and flew vertical spirals from ~300m to ~3500 m 275 over the cities of Julu, Quzhou, and Xingtai (Fig. 1). There were 11 research fights between May 8, 2016 and June 11, 2016. Wang et al. (2018) identified three different Planetary Boundary Layer (PBL) structures with distinct aerosol vertical structure. Their analysis of the aerosol pollution was mostly located below an altitude of 2 km, but sometimes with a vertically inhomogeneous structure, with higher aerosols at higher altitudes than at the surface but still in the boundary layer. 280 These vertical structures were mostly observed when the pollution originated from the southwest and from the eastern coastal region of the study domain, while cleaner air masses originated from the northwest. CO was measured by Cavity Ring Down Spectroscopy by the Picarro Model G2401-m instrument with a 5-second precision of 4 ppbv and an estimated accuracy of ±1% and $O_3$ by UV-absorption using a Thermal Electron Model 49C ozone analyzer. $O_3$ values ranged from 285 52 ppbv to 142 ppbv, partly because flight days were chosen to target meteorological conditions favorable to smog events (Benish et al., 2020). CO concentrations ranged 91 ppbv to about 2 ppmv (Benish et al., 2020). The pervasive high levels of CO correlated with $SO_2$ indicate extensive low-tech coal combustion. Therefore, we rejected individual CO observations (about 5% of total CO observations) when $SO_2$ was greater than 20 ppbv to remove the extremely polluted plumes.

**3 Model configuration and improvements**

### 3.1 Community Atmosphere Model with Chemistry (CAM-chem)

We use the open-source Community Earth System Model version 2.1 (CESM2.1); an overview of the modeling system and its evaluation is presented in Danabasoglu et al. (2020). It contains many new scientific features and capabilities, including an updated coupler, the Common Infrastructure 295 for Modeling the Earth (CIME), which allows for running an ensemble of CESM runs, in parallel with a single executable. The atmosphere is modeled using the finite volume dynamical core of the Community Atmosphere Model version 6 (CAM6) with 32 vertical levels and a model top at 3.6 hPa, and a 1.25° (in longitude) by 0.95° (in latitude) horizontal resolution (Gettelman et al., 2019). The model now uses a unified parameterization of the planetary boundary layer (PBL) and 300 shallow convection, the Cloud Layers Unified by Binormals (CLUBB, Bogenschutz et al. 2013). Other updates on the model physical parameterizations are described in Gettelman et al. (2019). The new Troposphere and Stratosphere (TS1) reduced gas phase chemical mechanism contains 221 species and 528 reactions (Emmons et al., 2020), and thus explicitly represents stratospheric and tropospheric ozone and OH chemistry. This chemical scheme contains many updates, 305 including on the isoprene oxidation mechanism, splitting a single aromatic into BENZENE, TOLUENE and XYLENES lumped species and a terpene speciation. The overall setup of CESM2.1 has been updated following the protocol of the Coupled Model Intercomparison Project Phase 6, which includes solar forcings (Matthes et al., 2017), surface greenhouse gases boundary conditions (Meinshausen et al., 2017) and anthropogenic emissions. Therefore, we use the 310 anthropogenic emission inventory of chemically reactive gases that has been generated by the Community Emissions Data System (CEDS, Hoesly et al., 2018). We use the latest year available (2014) for the KORUS-AQ period (2016). It is commonly acknowledged that errors in the emission inventory for China are much larger than the trends between different years (Feng et al.,





2020). Anthropogenic emissions over East Asia are replaced by the KORUS inventories version 5 or KORUS v5, based on the Comprehensive Regional Emissions for Atmospheric Transport Experiment (CREATE) (Woo et al., 2012). Daily Biomass Burning emissions are obtained from the Fire Inventory from NCAR (FINN v1.5) version 1.5 (Wiedinmyer et al., 2011). Biogenic emissions are modeled within the Community Land Model, thanks to the implementation of the Model of Emissions of Gases and Aerosols from Nature or MEGAN v2.1 (Guenther et al., 2012).

A summary of the model references is presented in Table 1.

**Table 1: Summary of the main model components and references for CESM2.1 / CAM6-Chem.**

| Model component | Reference |
| --- | --- |
| Community Earth System Model Version 2.1 (CESM2.1) | Danabasoglu et al., 2020 |
| Community Atmosphere Model version 6 (CAM6) | Gettelman et al., 2019 |
| Tropospheric and Stratospheric chemistry scheme (TS1) | Emmons et al., 2020 |
| Organic aerosol scheme (with Volatility Basis Set) | Tilmes et al., 2020 |
| Modal Aerosol Module (MAM4) | Liu et al., 2016 |
| Community Land Model (version 5) | Lawrence et al., 2019 |
| Model of Emissions of Gases and Aerosols from Nature (version 2.1) | Guenther et al., 2012 |
| Inputs | |
| Community Emissions Data System (CEDS) | Hoesly et al., 2018 |
| Comprehensive Regional Emissions for Atmospheric Transport Experiment (CREATE) version 5 or KORUS v5 | Woo et al., 2012 |
| Fire Inventory from NCAR (FINN v1.5) version 1.5 | Wiedinmyer et al., 2011 |
| Greenhouse gases prescribed fields | Meinshausen et al., 2017 |
| Methane net surface fluxes | Saunois et al., 2020 |

### 3.2 Sensitivity test on the biogenic emissions

The KORUS-AQ campaign was subject to photochemical episodes with large concentrations of secondary aerosols and ozone (e.g. Kim H. et al., 2018). There is a significant amount of biogenic emissions from the South Korean forests including deciduous oak trees (Lim et al., 2011) and conifers such as the Korean pine (Pinus koraiensis), both of which surround the Taehwa Research Forest site. As a result, there are large emissions from a variety of compounds, such as isoprene,

monoterpenes and sesquiterpenes, which contribute to enhanced ozone in favorable conditions (Kim S. Y. et al., 2013; Kim S. et al., 2015, 2016; Kim H.-K et al., 2018). Oak et al. (2020) showed that the largest ozone production efficiency was in the rural areas of South Korea, where biogenic emissions are dominant. Kim et al. (2014) studied how the Plant Functional Type (PFT) distributions affect the results of biogenic emission: broadleaf trees, needleleaf trees, shrub, and

herbaceous plants are significant contributors to BVOCs in South Korea. They found missing PFT data over Seoul and a large sensitivity in PFTs to changes in temperature, which could explain some uncertainties in the MEGAN model. We performed a sensitivity analysis to the biogenic emissions by increasing the emission factors for three of the Community Land Model PFT that are present in Korea, the "Needleleaf Evergreen Temperate Tree", the "Broadleaf Evergreen

Temperate Tree", and the "Broadleaf Deciduous Temperate Tree". We perform a set of simulations to determine the best fit to the observations of methanol ($CH_3OH$), ethene ($C_2H_4$), acetaldehyde ($CH_3CHO$), acetone ($CH_3COCH_3$), methyl hydroperoxide ($CH_3OOH$) and formaldehyde ($CH_2O$).


For the sake of clarity, we will present one experiment denoted as CAM-chem-Bio; more information can be found in the supplement.


## 4. Chemical data assimilation system

### 4.1 Data Assimilation Research Testbed (DART) implementation

The Data Assimilation Research Testbed (DART) is an open source community facility for ensemble data assimilation developed and maintained at the National Center for Atmospheric Research (Anderson et al., 2009a). DART has been used in numerous studies for Data Assimilation (DA) within CESM (Hurrell et al., 2012, Danabasoglu et al., 2020). Global DA analyses have been carried out with assimilation of conventional meteorological datasets within the Community
Atmosphere Model (CAM, Raeder et al. 2012), the Community Land Model version 4.5 or CLM4.5 (Fox et al. 2018), and in a weakly coupled atmospheric assimilation in CAM and oceanic assimilation in the Parallel Ocean Program ocean model (Karspeck et al. 2018). The Chemical Data Assimilation system inherits from previous work that coupled the Ensemble Adjustment Kalman Filter (EAKF) analysis algorithm (Anderson et al., 2001) with CAM-chem. The
DART/CAM-chem is designed for efficient ensemble data assimilation of chemical and meteorological observations at the global scale (Arellano et al., 2007; Barré et al., 2015; Gaubert et al., 2016, 2017).

### 4.2 DART/CAM-chem analysis and forecast algorithm


   The analysis is carried out using a deterministic ensemble square root filter, the Ensemble Adjustment Kalman Filter (EAKF) (Anderson 2001, 2003). The ensemble of 30 CAM-chem members is run with a single executable of CESM using the multi-instance capability. At the analysis step, the following model variables are updated when weather observations are
assimilated: surface pressure, temperature, wind components, specific humidity, cloud liquid water and cloud ice. Assimilated observations include radiosondes, Aircraft Communication, Addressing, and Reporting System (ACARS), but also remotely sensed data including satellite drift winds and Global Positioning System (GPS) Radio Occultation. We use a similar setup as previous studies (Barré et al., 2015; Gaubert et al., 2016, 2017) with a spatial localization of 0.1
radians or ~600 km in the horizontal and 200 hPa in the vertical for both chemical and meteorological observations. We now use the spatially and temporally varying adaptive inflation enhanced algorithm (El Gharamti 2018), that generalizes the scheme of Anderson (2009b). Multiplicative covariance inflation is applied to the forecast ensemble before each analysis step.

**4.3 MOPITT assimilation**

   As in previous implementations, both CO retrievals from MOPITT and meteorological observations are simultaneously assimilated within the DART framework. We assimilate profiles of retrieved CO from the MOPITT nadir sounding instrument onboard the NASA Terra satellite.
The MOPITT V8J product (Deeter et al., 2019) is a multispectral retrieval using the CO absorption in the Thermal Infra-Red (TIR, 4.7 μm) and Near Infra-Red (NIR, 2.3 μm) bands (Worden et al., 2010). The objective is to maximize the retrieval sensitivity to the lower layers of the atmosphere while minimizing the bias. We apply the same filtering thresholds that are used to create the L3 TIR-NIR product, which exclude all observations from Pixel 3 in addition to observations where
both (1) the 5A signal to noise ratio (SNR) is lower than 1000 and (2) the 6A SNR is lower than 400. We apply the strictest retrieval anomaly flags (all from 1 to 5). We only assimilate daytime





measurements, where latitudes are lower than 80 degrees and when the total column degrees of freedom are higher than 0.5. Super-observations are produced by applying an error-weighted average of the profiles (Barré et al., 2015) on the CAM-chem grid, with no error correlation since we consider those to be minimized by a strict use of the quality flags, as in Gaubert et al. (2016). In general, MOPITT data have errors smaller than 10% (Tang et al., 2020; Hedelius et al., 2019), which is much lower than model errors. We evaluate our assimilation results with fully independent aircraft observations.

**4.4 Ensemble design**

The ensemble of prior emissions is generated by applying a spatially and temporally correlated noise to the given prior emission field, as in previous studies (Gaubert et al., 2014, 2016, 2017; Barré et al., 2015, 2016). Emission perturbations are generated from a two-dimensional Gaussian distribution with zero mean and unitary variance (Evensen, 2003), with a fixed spatial correlation length. Here we applied the same set of perturbations for every time step, thus the prior ensemble has a temporal correlation of 1. A different noise distribution is drawn for Biomass Burning (BB) CO emissions than for anthropogenic direct CO emissions, with a decorrelation length of 250 km for BB, and 500 km for direct anthropogenic CO. Thus, as opposed to the previous studies, anthropogenic and BB CO sources are completely uncorrelated in the prior ensemble. The same noise is then applied to all the species emitted by the same source, BB or anthropogenic, including NMVOCs, the non-organic nitrogen species, $SO_2$, and aerosols. This means that emissions of NMVOCs and CO from the BB or anthropogenic sectors will be strongly correlated. We generated another noise sample with a decorrelation length of 500 km for soil emissions of NO.

The ensemble spread in the model physics variables is important for CO, which is directly sensitive to errors in horizontal and vertical winds (both boundary layer height and convection), as well as surface exchange, and indirectly through the impact of dynamics and physics on other chemicals. In particular, a spread in the MEGAN estimates of direct and indirect CO emissions from biogenic sources will be generated from the different atmospheric states passed to the land model. We assigned a spatially and temporally uniform noise drawn from a normal distribution with a standard deviation of 0.1 to the $CH_4$ emissions. More work will be done to generate a realistic spread in $CH_4$ emissions, but that is beyond the scope of this study.

**4.5 Variable localization and parameter estimation**

The multivariate error background error covariance allows for an estimation of the error correlation between the adjusted model variables or state vector and observations. As in previous studies we choose a strict "variable localization" (e.g. Kang et al., 2011), because (1) it is easier to quantify the impacts of the assimilation, such as the chemical response (Gaubert et al., 2016), as well as the model and observations errors (Gaubert et al., 2014); (2) spurious correlation can have a strong impact on the non-assimilated species that have no constraints.

In addition, several NMVOCs have been added to the state vector and consequently are also updated by the MOPITT observations. The NMVOCs with a strong anthropogenic and/or BB origin that have a primary sink with OH should be strongly correlated with CO (Miyazaki et al., 2012). The relationships between NMVOCs and CO leads to a correlation in their errors, so that the correlation existing in the ensemble will reflect those true errors. We added $C_2H_2$, $C_2H_4$, $C_2H_6$, $C_3H_8$, benzene, toluene, and the XYLENES, BIGENE and BIGALK surrogate species to the state vector. We also optimized the CO emissions from BB and anthropogenic sources separately. In





addition to the initial spread described above, spatially and temporally varying adaptive inflation
        is also applied to the optimized surface flux (SF) model variable during the analysis procedure.
        In CAM-chem, a diurnal profile is not applied to the emissions, instead emissions are interpolated
        from the dates provided in the inventories, which is daily for BB and monthly for anthropogenic
        sources. The relative increments obtained from the analysis in the form of the surface fluxes model
variable (SF) is propagated back to the input files emissions (E) following:

$$E_i^{analysis} = E_i^{prior}\left(1 + w\frac{\Delta SF_i}{SF_i}\right) \tag{4}$$

        where $i$ is an ensemble member and $w = a\,e^{-\frac{t}{\tau}}$ is a weight to represent the temporal
        representativeness and to limit the impact of spurious correlation. At the analysis time (t=0), the
weight will be w=a, with a=0.8, i.e. 80 % of the initial increments in Eq. 3. For the other time steps
        t, the exponential decay characteristic time, $\tau$, is set to 4 days in the case of BB and 4 months in
        the case of anthropogenic emissions. The impact of the increments will therefore decrease
        exponentially for the other time steps t from 0.8 to 0, which is imposed (bounded) for $2\tau$ (8 months
        or 8 days). This makes a strong correction for the current time and the closest time step. This allows
for smoothing the increments over time while hopefully leading to a convergence through the
        sequential correction of the emissions during the assimilation run.

### 4.6 Simulations overview

        In sections 5 and 6, two simulations with the assimilation of meteorological observations will be
presented, the Control-Run and the MOPITT-DA and the difference between the two simulations
        is the assimilation of MOPITT in the MOPITT-DA run. In the MOPITT-DA assimilation run, the
        initial conditions of CO and some NMVOCs, and CO emission inventories from anthropogenic
        and biomass burning sources, are optimized during the analysis step. The summary of the
        simulations presented in the following sections is presented in Table 2.
In section 6, additional sensitivity tests will be performed using deterministic CAM-chem
        simulations. In this case, since no meteorological data assimilation is performed, the dynamics
        from the prognostic variables U, V, and T need to be nudged towards the NASA GMAO
        GEOS5.12 meteorological analysis in order to reproduce the meteorological variability. The
        GEOS analysis is first regridded on the CAM-chem horizontal and vertical mesh. The nudging is
driven by two factors: the strength, a normalized coefficient that ranges between 0 and 1; and the
        frequency of the nudging, here configured to use 6-hourly outputs from either the GEOS5
        reanalysis or our own DART CAM-chem Control-Run. Based on an ensemble of sensitivity tests
        (SI), we use the nudging setup that minimizes the meteorological errors for the KORUS-AQ
        observations. This best simulation is the g-post-0.72, hereafter denoted as Prior (Table 2), and will
serve as a reference for the additional sensitivity simulation experiments. Aside from the Control-
        Run and the MOPITT-DA, the CAM-chem simulations have the same nudging setup, and only
        differ by the CO anthropogenic emissions flux. In addition, the CAM-chem-Bio is the same as the
        CAM-chem posterior but with an overall increase in the MEGAN emission factor.
        We compare our emission estimates with a state-of-the-art chemical data assimilation and
inversion system, the Tropospheric Chemistry Reanalysis version 2 or TCR-2 (Miyazaki et al.,
        2019b, Miyazaki et al., 2020b). They assimilate a variety of satellite instruments using the Local
        Ensemble Transform Kalman Filter (LETKF, Hunt et al. 2007) with the MIROC-chem model
        (Wanatabe et al. 2011). The setup is fully described and evaluated in Miyazaki et al. (2020b). We
        regridded the anthropogenic posterior CO estimate from their 1.125° × 1.125° mesh grid to the
CAM-chem grid and perform a simulation with anthropogenic CO and related tag from the TCR-





2 posterior emissions. In the TCR-2, the prior anthropogenic emission is HTAP v2 for 2010 (Janssens-Maenhout et al., 2015). Note that the simulations denoted as TCR-2 Prior and TCR-2 Posterior are CAM-Chem simulations with the respective anthropogenic CO emissions from TCR-2. We also use the Copernicus Atmosphere Monitoring Service (CAMS) global bottom-up emission inventory (Granier et al. 2019; Elguindi et al., 2020). We use the CAMS-GLOB-ANTv3.1, which has only minor changes with regards to the most recent version (v4.2). The gridded inventory is available at a spatial resolution of 0.1° × 0.1° and at a monthly temporal resolution for the years 2000-2020. It is built on the EDGARv4.3.2 annual emissions (Crippa et al., 2018) and extrapolated to the most current years using linear trends fit to the years 2011-2014 from the CEDS global inventory.

**Table 2: Summary of the simulations. The Nudging (GEOS) refers to a CAM-Chem deterministic runs with specified dynamics, using a nudging to GEOS-FP analysis winds and temperatures (see supplement). Aside from the DART simulations (first 2 rows), all the simulations have the same initial conditions and the same nudging and only change by their anthropogenic CO emissions inputs.**

| Simulation name | Meteorology | Emissions (prior) |
|---|---|---|
| Control-Run | Assimilation (DART) | Prior (CEDS-KORUS-v5) |
| MOPITT-DA | Assimilation (DART) | Optimized (CEDS-KORUS-v5) |
| Prior | Nudging (GEOS) | Prior (CEDS-KORUS-v5) |
| TCR-2-Prior | Nudging (GEOS) | Prior (HTAP v2) |
| Posterior | Nudging (GEOS) | Posterior (CEDS-KORUS-v5) |
| CAM-Chem-Bio | Nudging (GEOS) | Posterior (CEDS-KORUS-v5) + MEGANx2 (see SI) |
| TCR-2 | Nudging (GEOS) | Posterior (TCR-2, HTAP v2) |

## 5 Assimilation results: Evaluation of the posterior CO during KORUS-AQ

We use the fully independent DC-8 DACOM CO measurements to evaluate the MOPITT assimilation. Figure 2 compares the averaged vertical profiles for the 4 different mission weather regime phases (Peterson et al., 2019) and the average and standard deviation of all the flights. Observed background CO in the upper free troposphere is between 100 ppbv and 125 ppbv and show a variation of around 10 % between the different phases. The Control-Run shows an average background between 70 ppbv and 100 ppbv for the four phases and 80 ppbv for the full KORUS-AQ period, while the MOPITT-DA varies between 80 ppbv and 110 ppbv for the 4 phases with an average of 90 ppbv for the KORUS-AQ period. Because of the reduction of the CO in the middle troposphere (700 hPa to 300 hPa), the RMSE in MOPITT-DA is reduced by around 10 ppbv compared to the Control-Run.

For the layers closer to the surface, the temporal variations are much stronger. During Phase 3, observed CO is 44 % and 30 % higher than the campaign average at 850 hPa and 950 hPa, respectively. While this feature is much better reproduced after assimilation, absolute RMSE values remain large. Overall, the bias is greatly reduced for the MOPITT-DA in the layers between 850 hPa and 650 hPa. We note that the mean CO is still lower than the average observations. The MOPITT-DA shows at the 950 hPa and 850 hPa levels an underestimation of around 30 ppbv, i.e. between 10 % and 20 % lower than the observations. This is in the range of the expected performance given the retrieval uncertainties (10 %) and the spatial footprints of MOPITT pixels 22km x 22km)



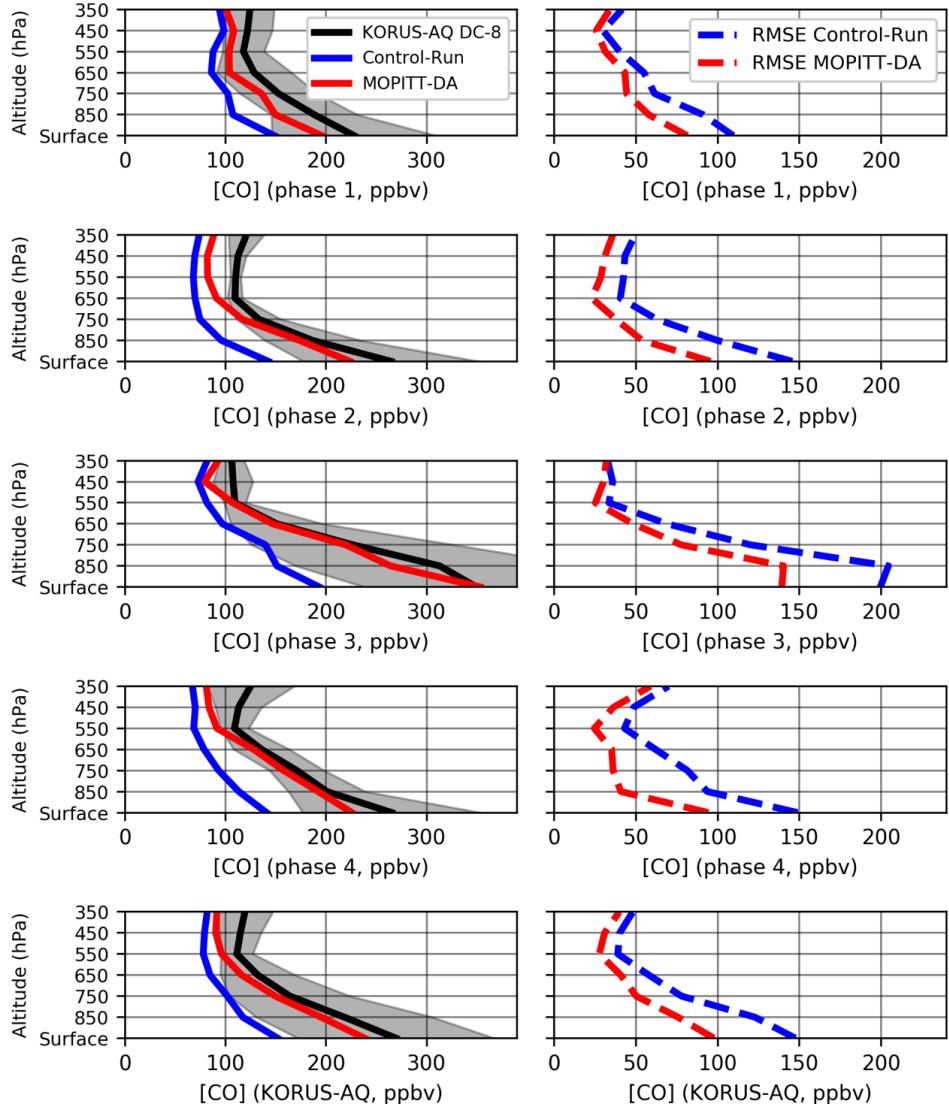

525

**Figure 2: Average CO profiles (left panels) and related RMSE (right panels) for the Control-Run and the MOPITT-DA. The mean (black line) and standard deviation (shaded grey) of the DC-8 observations are calculated for each 100 hPa bin. The first 4 rows are averaged over the different weather regimes of the campaign (Peterson et al. 2019). The last row displays the average over the whole campaign.**

530

## 5.1 VOCs state vector augmentation

Concentrations of some VOCs have been added to the state vector and are therefore optimized, according to the covariance estimated by the ensemble, when MOPITT observations are assimilated. This means that if true errors in the VOCs are not correlated to CO, only noise will be
535  introduced in the ensemble of chemicals. However, if there is a correlation between the modeled CO state and VOCs, then errors could be reduced, even though VOCs are not directly observed.





**Table 3: VOCs added to the state vector, corresponding measuring instrument, and lifetime (Simpson et al., 2020) used for validation. For comparison with surrogate species, the sum of all the corresponding VOC observations is used. WAS stands for Whole Air Sampler.**

| Model Variable | Observations | Lifetime (days) |
|---|---|---|
| $C_2H_6$ | Ethane (WAS) | 48 |
| $C_2H_2$ | Ethyne (WAS) | 15 |
| $C_3H_8$ | Propane (WAS) | 11 |
| BENZENE | Benzene (PTRMS) | 9.5 |
| BIGALK | i-Butane, n-Butane, i-Pentane, n-Pentane, n-Hexane, n-Heptane, n-Octane, n-Nonane, n-Decane (WAS) | 3.5 +/- 1.6 |
| TOLUENE | Toluene (PTRMS) | 2.1 |
| $C_2H_4$ | Ethene (WAS) | 1.5 |
| XYLENES | mp-Xylene, o-Xylene (WAS) | 0.7 +/- 0.2 |
| BIGENE | 1-Butene, i-butene, trans-2-Butene, cis-2-Butene, 1-3-Butadiene (WAS) | 0.2 +/- 0.1 |

The list of optimized VOCs is shown in Table 3, together with their lifetime and the corresponding species from the Whole Air Sampler (WAS) instrument used for evaluation. An increase in concentration is found for all 9 VOCs in the MOPITT-DA simulation, either because of the state augmentation, or because of the reduction in OH due to CO adjustments. Even if the changes are small, this can lead to an increase in errors for the vertical profiles compared to observations when the species is already overestimated in the lower layer of the atmosphere. This is the case for $C_2H_4$ and BIGENE, the only two species that have substantial biogenic and fire sources, as well as for xylenes and toluene. For all the other species, which are underestimated and are mostly from anthropogenic sources, the assimilation leads to an improvement compared to the observations, mostly by reducing their biases. The best results are obtained for ethane and to a lesser extent propane (Fig. 3). Despite the broad anthropogenic source, ethane and propane originate from sectors that are quite different from CO. However, CO, ethane and propane have one thing in common which is that their major atmospheric sink is through OH oxidation. This suggests that a bias in OH leads to correlated errors between CO and alkanes that can be mitigated by including these species to the state vector. We define a metric of improvement based on the relative change in RMSE that is positive when the RMSE is reduced. Figure 3 shows a clear dependence of this metric on the atmospheric lifetime of the VOCs. All the modeled VOCs with a lifetime shorter than 5 days show an increase in errors, while all the VOCs with a lifetime greater than 10 days are improved, with the largest improvement for ethane, which has a lifetime of 48 days. The relatively large spatial and temporal scales of CO that arise due to its medium atmospheric lifetime significantly limit the ability of CO assimilation to resolve the high-frequency changes in those compounds with short lifetimes. More importantly, this is also to be expected given the limited sensitivity of the MOPITT observations to boundary layer CO.

While satellite observation spatiotemporal resolution and sampling might be improved in the future, NMVOCs with a lifetime shorter than several days should not be included in the state vector when assimilating CO. However, the concentrations of NMVOCs with strong anthropogenic or BB sources and similar chemical characteristics to CO might be significantly improved by the assimilation. We believe that this could also be true for methane.

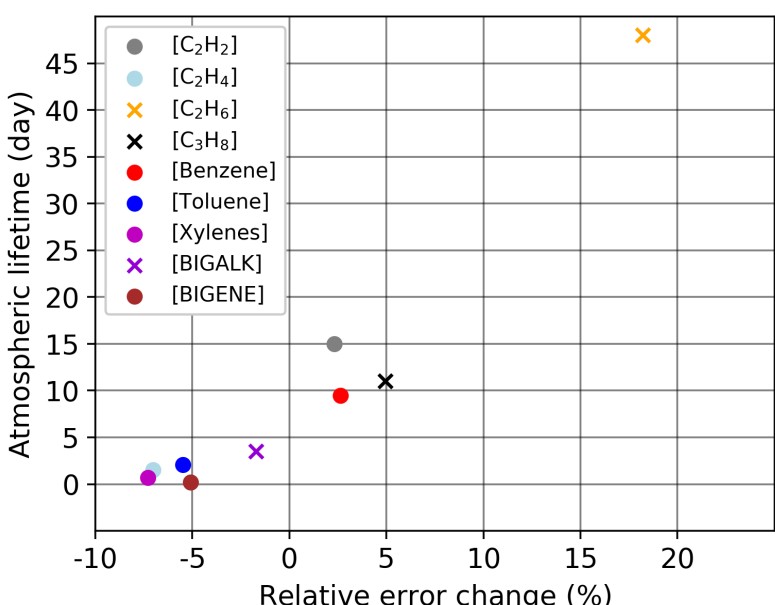

**Figure 3: Atmospheric lifetime (from Simpson et al., 2020) in days for the VOCs added to the state vector. Xylenes, BIGALK and BIGENES are surrogate species, so an average of the lifetimes is calculated. The relative error change is the opposite of the difference in Root Mean Square Error relative to the control. Thus, a positive relative error change means an improvement compare to the control run.**

## 5.2 Chemical response from MOPITT-DA

This section presents a short summary of the impact of the CO assimilation on the chemical state of the atmosphere and the comparison with unobserved species. Figure 4 shows the average vertical profiles for OH, $HO_2$, $CH_2O$ and $O_3$. We use simulated OH and $HO_2$ from the observationally constrained NASA LaRC box model (Schroeder et al., 2020). The Control-Run and the LaRC box models agree on the mean OH spanning the first two binned layers, at lower altitudes. Aloft, the Control-Run overestimates the LaRC box model simulations. The Control-Run underestimates $HO_2$, which suggests that the excellent agreement on OH in the boundary layer is likely caused by compensating errors. That is, the increase of CO through the MOPITT assimilation decreases the OH concentrations (Gaubert et al. 2016). Here, we find better agreement of the model OH with the observationally constrained LaRC box model simulation at 750 hPa and above. This in turn increases $HO_2$ and shows a better match with the LaRC box model. This suggests that the $HO_2$ underestimation can be partly explained by the CO underestimation.

Overall, the sum of OH and $HO_2$ is underestimated, and therefore the CO chemical sink alone cannot explain the CO underestimation during the campaign. Alternatively, $CH_2O$ is underestimated in both simulations, suggesting an underprediction of the secondary CO. A similar effect to that described in Gaubert et al. (2016) is shown, where an increase in CO through the sequential assimilation leads to reduced formaldehyde formation, albeit a small effect. In the lower part of the atmosphere, the oxidation of extra CO leads to more effective ozone production and no changes above, consistent with observations. The low $CH_2O$ and $O_3$ points to a missing NMVOC source. The comparison with NO, $NO_2$, $HNO_3$, $J(O_3)$, $J(NO_2)$ and $H_2O_2$ and PAN (shown in Figure S2 of the Supplement) suggests that while the $NO_x$ levels are reasonably simulated, $J(NO_2)$ is





underestimated and $HNO_3$ overestimated, which might explain the $HO_x$ partitioning. A lower value
       of the $HO_2$ heterogeneous uptake coefficient than the one used here (γ=0.1) might produce better
       results by reducing the $HO_2$ sink (see Appendix A2).

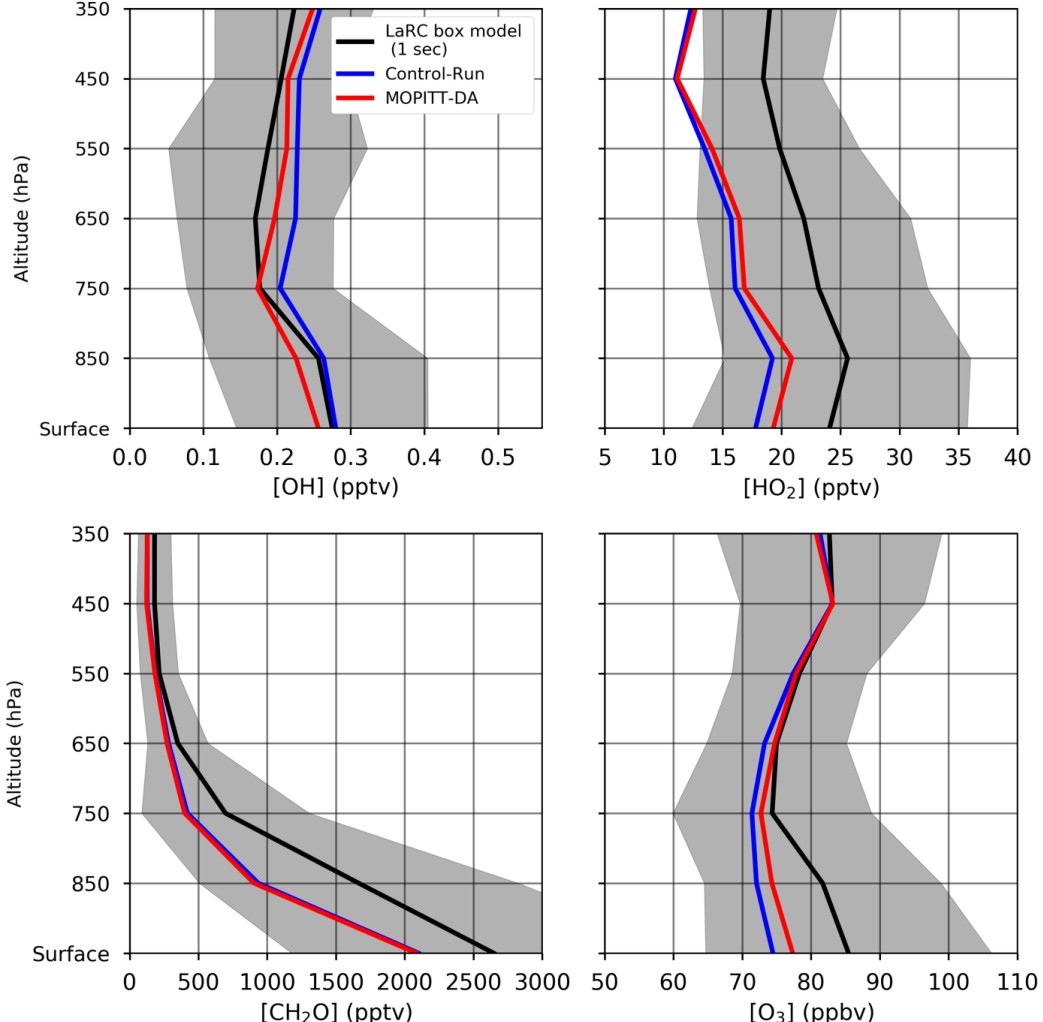

**Figure 4: Average vertical profiles of OH (top left) HO₂ (top right) for the 1-sec merge and the LaRC box model**
       **estimates (Schroeder et al., 2020). Results are shown for DC-8 1-min merge for CH₂O (bottom left) and O₃**
       **measurements (bottom right). The grey shaded area corresponds to the standard-deviation around the**
       **observed mean.**

**6 Anthropogenic CO Emissions, evaluation during ARIAs and KORUS-AQ**

       This section presents the sensitivities to the choice of anthropogenic CO inventories and to
       biogenic NMVOCs emissions. We further evaluate the simulated profiles of CO, $O_3$, OH, and $HO_2$
       with the observations from ARIAs and KORUS-AQ.

### 6.1 Comparison of anthropogenic emission estimates

We show in Figure 5 the emissions of the prior (CEDS-KORUSv5), and its posterior, estimated through the DART/CAM-chem inversion. It also shows the prior (HTAP v2) from the TCR-2 and its posterior estimate, for which CO emissions are also constrained by MOPITT. We also show the CAMS emissions.

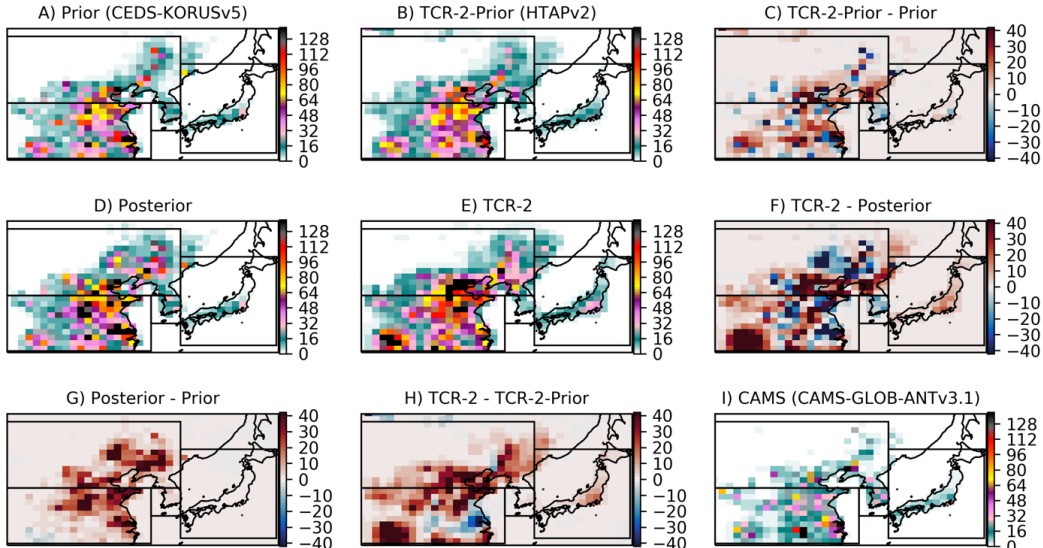


**Figure 5: Emissions flux for May 2016 in MgCO.month⁻¹. Prior (A, CEDS-KORUS v5), TCR-2-Prior (B, HTAP v2) and the difference between the 2 priors (C, TCR-2-Prior – Prior). The second row shows the Posterior (D, estimated by DART/CAM-Chem), the TCR-2 (E) and the difference between the 2 posteriors. The last row shows the emissions increments, the difference between the Posterior and the Prior (G) and between TCR-2 and TCR-2-Prior (H). The CAMS emissions are shown on the last panel (I).**


Compared to the prior (Fig. 5a), our posterior estimate (Fig. 5d) shows a reduction around the Guizhou province, in southwest China. Larger changes are observed for the Shandong and Henan provinces in central China and over the Yangtze River Delta (Fig. 5g). Increases in emissions are
also large in the North China Plain and the Liaoning Province. While both inversions show large increase over northern China (Fig. 5g, h), the spatial patterns of the emissions are different between the posterior and the TCR-2 for northern China. The TCR-2 emission increments are located more in the North China Plains and North Korea (Fig. 5f). Large differences can be identified in central China. The Shanghai megacity emissions are higher in the DART/CAM-chem posterior (Fig. 5d)
and the TCR-2 prior (Fig. 5b) than in the TCR-2 posterior (Fig. 5e). A more consistent pattern of larger emissions in the TCR-2 compared to our posterior is found in southern China and the Sichuan province (Fig. 5h and Fig. 5f). In the absence of a model for anthropogenic emissions, the spatial patterns of the prior emissions are important and likely to explain the difference between the TCR-2 and our DART/CAM-chem estimate. Another important aspect is the 500 km
correlation length initial perturbation to generate the ensemble of anthropogenic emissions, combined with a similar localization radius of ~600 km, which explains the large-scale increments found in the DART/CAM-chem emissions increments (Fig 6g). The TCR-2 prior show more emissions over North Korea than South Korea (Fig 6b) and the opposite is true for the DART/CAM-Chem prior (Fig 6c). This is reflected in the posterior where the TCR-2 has more





emissions in North Korea than the DART/CAM-Chem posterior (Fig. 5f). Compare to its prior, the DART/CAM-Chem posterior emissions are increased by 25 % for South Korea, with a larger change over the SMA (Fig. 5g). While the CAMS emissions are generally lower (Fig. 5i), the South Korean emissions are larger than in all the other inventories.

Our assimilation suggests an overall underestimation of bottom-up emission inventories for China (Fig. 6), consistent with previous studies. The agreement between the posterior emissions for China is better than for the bottom-up (Fig. 6). The difference between CAMS (3.65 TgCO.Month$^{-1}$) and the CEDS-KORUSv5 is (5.7 TgCO.Month$^{-1}$) twice as high as is two times higher than the difference between DART/CAM-chem posterior (7.6 TgCO.Month$^{-1}$) and TCR-2 (8.7

TgCO.Month$^{-1}$). On average, the increase in emissions due to assimilation is about 33 % for central China and nearly doubled (80 %) in Northern China, from 2.7 TgCO.Month$^{-1}$ to 4.9 TgCO.Month$^{-1}$. TCR-2 suggests higher emissions (5.7 TgCO.Month$^{-1}$), while the CAMS estimate is lower (1.8 TgCO.Month$^{-1}$). More work should be dedicated to improving the regional distribution and scaling up the baseline emissions.


For South Korea, a relatively smaller difference between the posterior than for the prior suggests an improved bottom-up inventory. However, the smaller area of South Korea is much less constrained by MOPITT, and the overall estimate seems to be determined by the prior distribution. For instance, the TCR-2 shows larger emissions over North Korea and the Pyongyang area while

DART/CAM-chem and CAMS suggests larger emissions for the SMA. Therefore, the CAMS total emissions that show a similar pattern (0.18 TgCO.Month$^{-1}$) are in good agreement with the DART/CAM-chem (0.16 TgCO.Month$^{-1}$) while the TCR-2 has a total of 0.07 TgCO.Month$^{-1}$. For Japan, where biomass burning and low-tech coal combustion are rare, the total is nearly unchanged in contrast to the other regions, and emissions are increased from 0.38 to 0.41 TgCO.Month$^{-1}$ or 8

670  %.





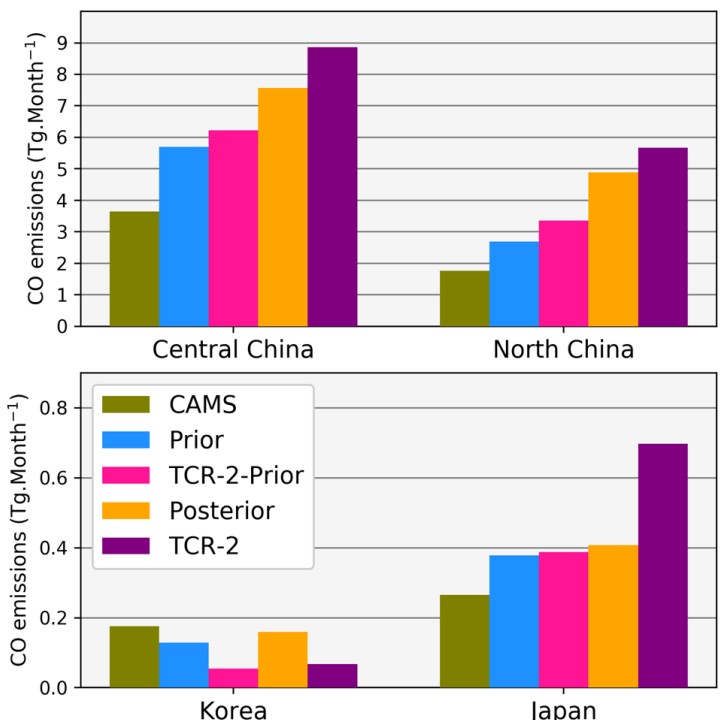

**Figure 6: Anthropogenic CO emissions for May 2016 for Central China (91E, 29N to 124E, 38N), North China (91E, 38N to 130E, 49N), South Korea (125E, 33.5N to 129E, 38N) and Japan (130E, 30N to 146E, 44N).**

## 6.2 Evaluation of the simulated vertical profiles against ARIAs and KORUS-AQ

### 6.2.1 Mean profile during ARIAs and KORUS-AQ

Figure 7 shows the average CO vertical profiles for the ARIAs and the KORUS-AQ campaigns. For the ARIAs field campaign, we bin the profiles into 50 hPa bins. Overall, CO observations show a strong variability, with large enhancement over a background of around 170 ppbv found at 775 hPa and above. Benish et al. (2020) show that the median of the observed CO values in the

lowest 500 meters is around 400 ppbv. Using additional enhancement ratios, the measurements indicate low-efficiency fossil fuel combustion, that could originate from residential coal burning and gasoline vehicles as well as crop residue burning such as straw from winter wheat. The MOPITT-DA and the TCR-2 overestimate the CO concentrations compared to the measurements for this surface layer although this overestimate is smaller when a value higher than 20 ppbv $SO_2$

(the approximate 95[th] percentile) is used to define plumes for exclusion. The CAM-chem posterior simulated CO concentrations, that just use the smoothed posterior emissions from the MOPITT-DA have a mean value close to the observations. Interestingly, the HTAP v2 inventory that was for the year 2010 still provides good CO profiles (TCR-2-Prior). The CAM-chem prior, a nudged CAM-chem simulation, and the Control-Run, underestimate CO concentration, with slight

differences due to transport. The modeled profile with CAMS emissions profiles is the lowest CO of all simulations. For altitudes above 900 hPa the bias is lowest using the TCR-2 emissions or with the MOPITT-DA, because these emissions are more spatially representative of regional pollution (Wang et al., 2018). This confirms that the free-tropospheric background is too low when





using CEDS-KORUSv5 and CAMS emissions. The MOPITT-DA naturally shows the lowest bias in CO concentrations in the free troposphere. The layer mean (and median) observed ozone during ARIAs (Benish et al., 2020) is around 80 to 90 ppbv and the mean peaks at 90 ppbv at 875 hPa (900 to 850 hPa). For this layer, simulations with higher CO are better reproduce the large $O_3$ peak. It means that the CO acts as a substantive precursor for ozone production, and is more consistent with the observations. The mean $O_3$ concentration is still underestimated by around 10 ppbv in the
free troposphere.

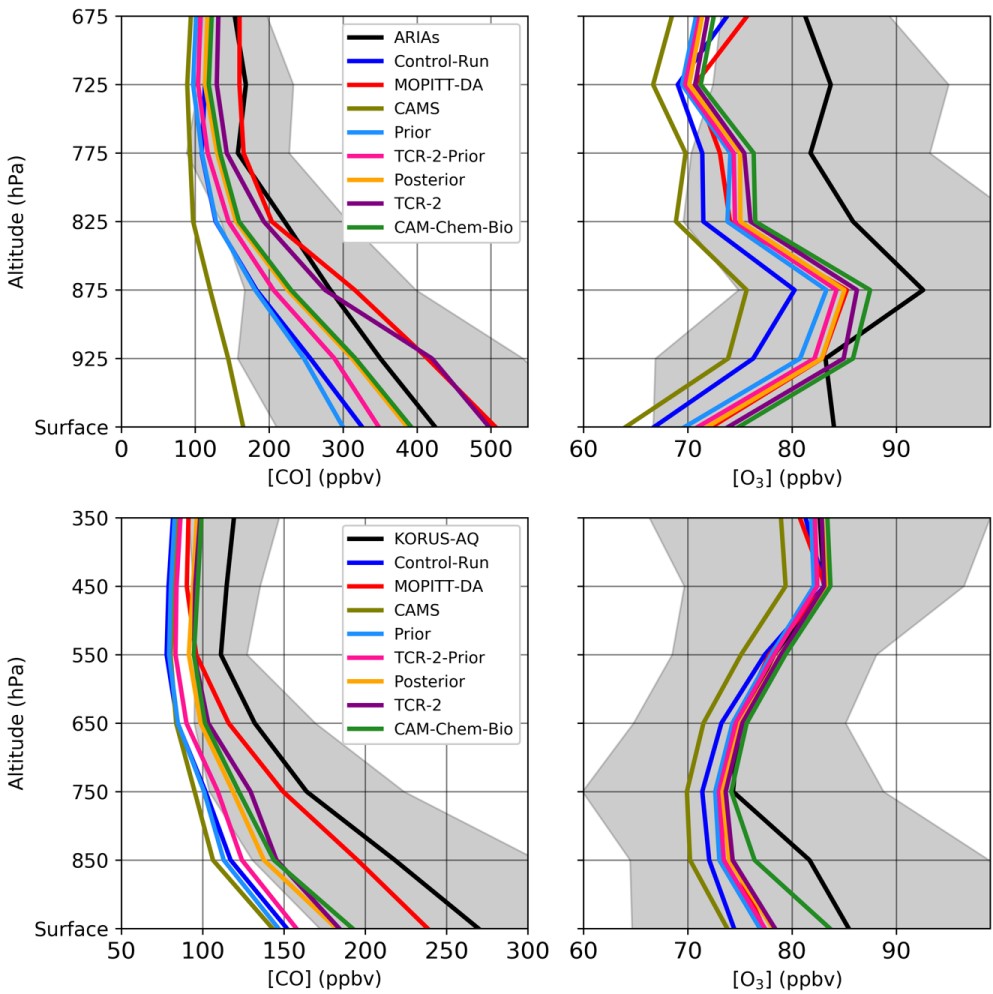

**Figure 7: Average CO vertical profiles for the ARIAs campaign (top panel) and the KORUS-AQ (bottom panel). Observations were filtered out when $SO_2$ was higher than 20 ppbv for ARIAs and benzene higher than 1 ppbv for KORUS-AQ. The black line shows the observation mean and the shaded area is the observation**
**standard deviation. Only mean CO or $O_3$, are shown for the model simulations.**

Two groups appear when comparing to KORUS-AQ observations. The prior group, using CAMS and CEDS-KORUSv5 with two different model dynamics, are simulating a lower CO and show a severe low bias of more than 100 ppbv at the surface. The second group includes the CAM-chem simulations using the DART/CAM-Chem emissions or the TCR-2 emissions. Those simulations
are quite close together with an average CO of 141 ppbv and 145 ppbv with a bias of 31 % and 29





% respectively (Table 4). This is to be compared with their priors that have an average CO of 116 ppbv and 125 ppbv, which means an underestimation of 43 % and 39 %, respectively. Correcting only the bias in anthropogenic emissions is not as efficient as the joint optimization of anthropogenic emissions and sequential optimization of initial conditions through data assimilation (MOPITT-DA). The MOPITT-DA has an average CO of 179 ppbv, resulting in 12 % underestimation on average (Table 4). It means other confounding factors can play a role, such as errors in transport or chemistry. Aside from CAMS, the modeled free tropospheric $O_3$ shows no particular bias. The enhancement of observed $O_3$ closer to the surface is underestimated in all simulations. The optimized emissions lead to an increase of a few ppb in $O_3$, bringing those simulations closer to the observations. In summary, using top-down estimates of CO emissions clearly improves the CO and $O_3$ vertical profiles against independent observations over China and Korea.

**Table 4: Comparison of CO (ppbv) measured aboard the DC-8 and model simulation for all altitudes. Statistical indicators are calculated for phase 1 (7 flight days, 2952 observations), phase 2 (4 flight days, 2029 observations), phase 3 (3 flight days, 1243 observations), phase 4 (5 flight days, 2448 observations) and the whole campaign (20 flight days, 9099 observations).**

|  | CO (1) | Bias (%) | CO (2) | Bias (%) | CO (3) | Bias (%) | CO (4) | Bias (%) | CO (All) | Bias (%) |
|---|---|---|---|---|---|---|---|---|---|---|
| **Observation** | 173.1 |  | 198.3 |  | 246.8 |  | 211.2 |  | 203.6 |  |
| **Control-Run** | 114.5 | -33.8 | 108.6 | -45.2 | 138.7 | -43.8 | 115.3 | -45.4 | 118.6 | -41.8 |
| **MOPITT-DA** | 146.5 | -15.4 | 168.3 | -15.1 | 230.6 | -6.6 | 182.8 | -13.5 | 178.5 | -12.4 |
| **CAMS** | 108.1 | -37.6 | 110.4 | -44.3 | 112.2 | -54.5 | 119 | -43.6 | 112.8 | -44.6 |
| **Prior** | 112.3 | -35.1 | 110.8 | -44.1 | 124.7 | -49.5 | 115.7 | -45.2 | 115.9 | -43.1 |
| **TCR-2-Prior** | 118.7 | -31.4 | 115.3 | -41.8 | 137.3 | -44.4 | 128.5 | -39.2 | 124.6 | -38.8 |
| **Posterior** | 136 | -21.4 | 131.8 | -33.5 | 157.1 | -36.3 | 139.5 | -33.9 | 140.9 | -30.8 |
| **TCR-2** | 138.4 | -20 | 128.9 | -35 | 174.4 | -29.3 | 146.1 | -30.8 | 145 | -28.8 |
| **CAM-Chem-Bio** | 138.4 | -20.1 | 137.2 | -30.8 | 163 | -34 | 151.8 | -28.1 | 147.2 | -27.7 |

### 6.2.2 Weather induced dynamical change in CO during KORUS-AQ

Figure 8 shows the CO anomalies during KORUS-AQ for the observations and the simulations. The CO anomalies are largest in phase 3, with an enhancement of almost 100 ppb at 850 hPa. This transport phase, defined and described in Peterson et al. (2019) was characterized by high levels of ozone (>60 ppbv) and $PM_{25}$ (>50 μg/m³) because of efficient transport of low-level pollution (Huang et al., 2018; Miyazaki et al., 2019; Choi et al., 2019). The model reasonably reproduced the variability of the different phases, albeit with insufficient magnitude. The desired magnitude is only achieved when including data assimilation. Updating the anthropogenic emissions from the bottom-up to the top-down inventories improved the representation of the CO anomalies. This suggests that weather patterns and the direct anthropogenic emissions explain most of the CO variability during the campaign. Large-scale subsidence and reduced wind speeds during the anticyclone of phase 2 were marked by the lowest CO anomalies and are also better reproduced with the updated emissions. Over South Korea, running CAM-chem with the CAMS emissions shows the largest anthropogenic CO from South Korean sources at the surface for the 4 phases and is likely to produce more realistic simulation since CO is constantly underestimated. This cannot be seen for the total CO since most of the CO is not from South Korean direct anthropogenic



sources. The profile tags of the contributions from Central China and Northern China are approximately doubled with the optimized emissions, consistent with Tang et al. (2019). As shown in the previous section, the TCR-2 and the DART/CAM-chem posteriors have the largest emissions from China and therefore the largest contribution of the CO tags from both Northern and Central China.

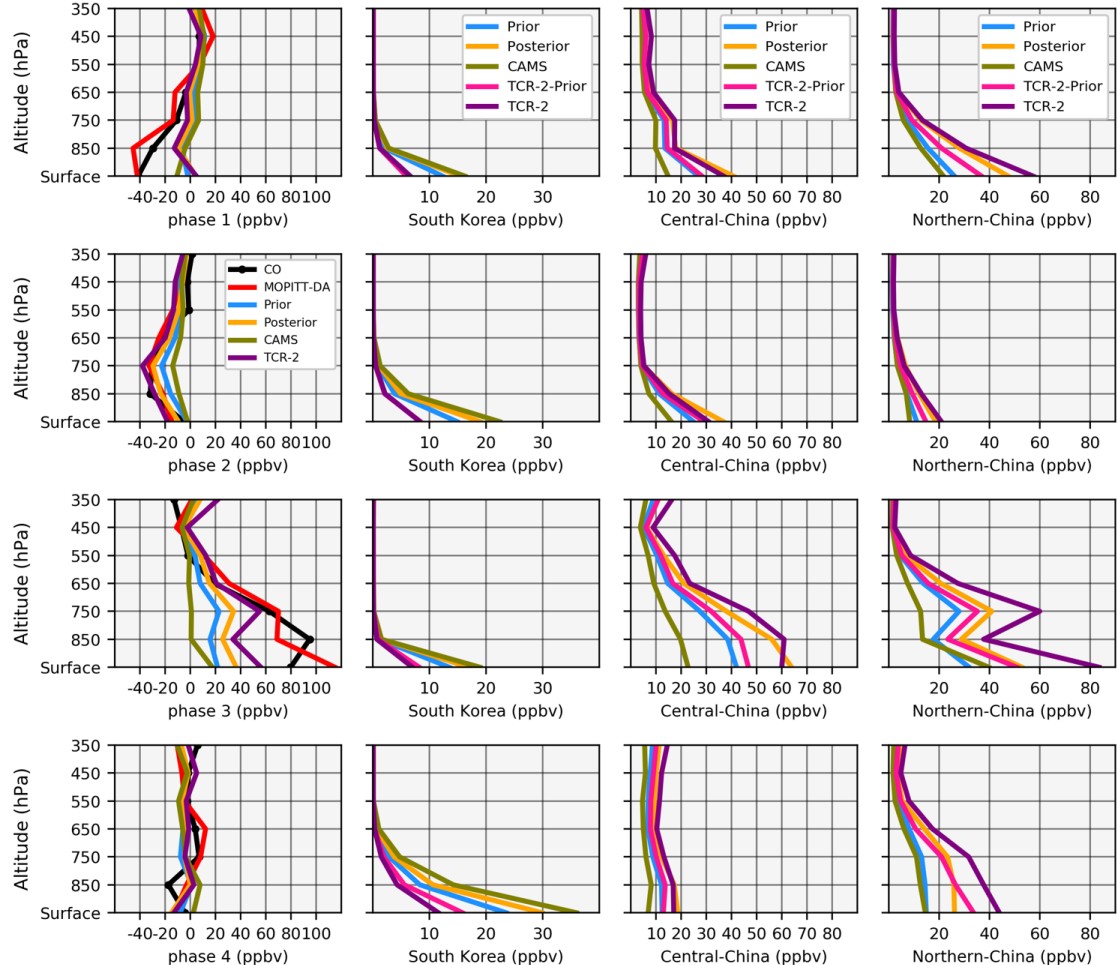

**Figure 8: Average CO anomalies for the four different phases of KORUS-AQ (first column). The anomaly is defined by subtracting the respective average vertical profile (see Fig. 2). Absolute vertical profiles of the CO tags are shown from South Korea (Second column), Central China (third column) and Northern China (fourth column). Each row corresponds to a different phase.**

We will now focus on two case studies, Phase 2 and Phase 3, for which the highest ozone was observed at the surface in South Korea during KORUS-AQ (Peterson et al., 2019).

### 6.2.3 Phase 2 case study: the anticyclonic phase

A large-scale anticyclone occurred from 17 May 2016 to 22 May 2016 with increased surface temperatures, reduced wind speed and drier conditions, all of which enhance ozone production (Peterson et al., 2019). The conditions were also favorable for an increase in biogenic emissions.



As shown in the previous sections, this episode was characterized by negative CO anomalies that
were best captured by the MOPITT-DA simulations. This anomaly is reflected through the OH
(and $O_3$) vertical profiles that also follow lower (higher) concentrations for higher altitudes,
respectively (Figure 9). This indicates rather clean air masses, probably from stratospheric origin.
This episode is driven by the overall weather pattern with a clear enhancement of $HO_x$ and $O_3$
towards the surface. In this case, changes in the anthropogenic CO only play a minor role, still the
$O_3$ is modeled better with a reduction of the bias by 1 ppbv between the posterior and the prior
(Table 5). The increase in biogenic emissions leads to an improvement in $O_3$ by further reducing
the bias at the surface (Figure 9). Over the whole profile, the bias is reduced by 3 ppbv (4 ppbv
against the prior) for the CAM-Chem-Bio, compared to the CAM-Chem-Posterior, with a
reduction in RMSE as well (Table 5).



Figure 9: Average LaRC box model OH and HO₂ and measured O₃ for phase 2 (left column) and phase 3 (right column) of KORUS-AQ.

### 6.2.4 Phase 3 case study: low-level transport and haze development.

Phase 3 was characterized by the largest observed CO and O₃ positive anomalies. In this case, there is a clear relationship between the CO bias, and the O₃, OH and HO₂ vertical profiles (Figure 9).





The OH is overestimated because of a lack of CO and other VOCs. Increasing CO in the CAM-Chem posterior reduces OH and increases $HO_2$ and $O_3$. The overall bias (Table 5) in ozone is reduced from 11.3 ppbv to 9.9 ppbv with the change in CO, and lowered further to 7.3 ppbv with the additional increase in biogenic emissions (CAM-Chem-Bio). The relative impacts of biogenics are clear in the surface layer for OH, $HO_2$ and $O_3$. Overall, $HO_2$ and $O_3$ are underestimated as a result of CO underestimation. The MOPITT assimilation provides the best results for OH throughout the profile and lower RMSE and a similar bias as the CAM-Chem-Bio (Table 5). As suggested by the Chinese origin of the pollution for higher levels, it is likely that additional anthropogenic NMVOCs are also missing and contribute to the ozone formation that is still underestimated.

**Table 5: Comparison of O₃ measured aboard the DC-8 and model simulation for all altitudes. Statistical indicators are calculated for phase 2 (4 flight days, 1910 observations), phase 3 (3 flight days 1111 observations) and all KORUS-AQ.**

| | $O_3$ (Phase 2) | Bias | RMSE | $O_3$ (Phase 3) | Bias | RMSE | $O_3$ (All) | Bias | RMSE |
|---|---|---|---|---|---|---|---|---|---|
| Observation | 87.7 | | | 91.5 | | | 82.1 | | |
| Control-Run | 77.2 | -10.6 | 19.6 | 81.8 | -9.7 | 20.4 | 75.1 | -7 | 16.8 |
| MOPITT-DA | 79.7 | -8 | 18.1 | 83.8 | -7.7 | 19.5 | 76.9 | -5.2 | 15.9 |
| CAMS | 74.4 | -13.3 | 20.7 | 76.1 | -15.4 | 24.2 | 73.6 | -8.5 | 18.8 |
| Prior | 78.2 | -9.6 | 18.5 | 80.2 | -11.3 | 21.5 | 76.5 | -5.6 | 17.4 |
| TCR-2-Prior | 78.5 | -9.2 | 18.4 | 80.7 | -10.8 | 21.2 | 76.9 | -5.2 | 17.3 |
| Posterior | 79.1 | -8.6 | 18.2 | 81.6 | -9.9 | 20.8 | 77.6 | -4.5 | 17.2 |
| TCR-2 | 79.2 | -8.6 | 18.1 | 82.2 | -9.2 | 20.5 | 77.8 | -4.4 | 17.1 |
| CAM-Chem-Bio | 82.3 | -5.5 | 15.9 | 84.2 | -7.3 | 20.2 | 80.5 | -1.6 | 16.5 |

## 7 Conclusions

Anthropogenic CO emissions are an important contributor to poor summer air quality in Asia and to forward modelling uncertainties. Here we evaluate top-down estimates of the CO emissions in East Asia with aircraft observations from two extensive field campaigns. There are multiple lines of evidence that the bottom-up anthropogenic emissions are too low in winter and spring, leading to a large underestimation of CO during the KORUS-AQ campaign in May and June 2016. We also highlight in this work that chemical production and loss via OH reaction from emissions of anthropogenic and biogenic VOCs confound the attribution of this bias in current model simulations. Combined initial conditions and emission optimization remains the best method to overcome these modeling issues. The major findings of this investigation are:

1. The comparison of OH modeling and observations confirms that assimilating CO improves the OH chemistry by correcting the OH/$HO_2$ partitioning. The interactive and moderately comprehensive chemistry with resolved weather from reanalysis datasets represents well the variations in OH. These results provide an additional line of evidence that assimilating CO improves the representation of OH in global chemical-transport models. This has implications for studying the $CH_4$-CO-OH coupled reactions and the impact of chemistry and interactive chemistry for allowing feedbacks. It suggests that even if global mean OH is buffered on the global scale, local changes in OH can be important, and can be quantified by taking advantage of field campaigns. This will provide ways to improve and provide additional constrain on $CH_4$ inversions by either improving the sink or by better characterizing anthropogenic sources through CO assimilations. A better quantification of


the spatio-temporal variability of these compounds will improve the physical representation of Earth system processes and feedbacks and will be beneficial for both air quality and climate change mitigation scenarios.

2. The setup of the CO assimilation that corrects the initial conditions and emissions provides the best results for CO. While the emission update improves the forecast closer to the source, the assimilation allows for better reproduction of the vertical profiles and the background and eventually compensates for model errors.

3. The spread of emission estimates from state-of-the-art inventories, 3 bottom-up and 2 top-down is significant. For example, the emissions of Central China show a range from 3.65 $TgCO.month^{-1}$ to 8.87 $TgCO.month^{-1}$. Inventories with the highest emissions fluxes show improved vertical profiles of CO.

4. Running the forward model with updated emissions of anthropogenic CO increases the $O_3$ formation, reduces OH and increases $HO_2$. This improves the comparison with $O_3$, OH and $HO_2$ observations. The comparison with observations suggests that the overall modeled photochemistry was improved with updated CO emissions. In this case, there is also a better representation of severe pollution episodes with large $O_3$ values. Often overlooked, it clearly shows that running chemistry transport models with biased CO and VOCs emissions results in poorly modeled ozone and impacts most of the chemical state of the atmosphere. The sensitivities may vary for different chemical and physical atmospheric environments. In this case, underestimating CO in VOC-limited chemical regimes explains the underestimation of ozone in the boundary layer and the lower free troposphere.

5. Biogenic emissions appear to play an important role in ozone formation over South Korea, in particular when conditions are favorable (sunny and warm). The role is weaker over China, at least in May before maximum biogenic emission rates. A combined assimilation of CO and $CH_2O$ observations is likely to greatly improve ozone forecasting through estimates of boundary and initial condition estimates of VOCs.

On top of CO data assimilation, improved emissions through state augmentation can help improve the next-generation of Korean (e.g., Lee et al. 2020) or global (Barré et al., 2019) air quality analysis and forecasting systems. Further improvements can be achieved by simultaneously assimilating $CH_2O$ retrievals (e.g. Souri et al., 2020) and CO retrievals. Improving the aerosol distribution can help correct the $HO_2$ uptake and therefore OH, CO and $O_3$ by assimilating satellite aerosol optical depth measurements, in particular for this region with high aerosol loadings (e.g. Ha et al., 2020). Using CrIS-TROPOMI joint retrievals (Fu et al. 2016), the improved vertical sensitivity may potentially be used to further constrain secondary CO formed through biogenic oxidation. In this case, secondary CO is correlated with ozone formation. This is also true for other geographical areas, such as over the United States in summer (Cheng et al., 2017, 2018). On average, there is a lower combustion efficiency in China than in Korea, with the ratio of CO to $CO_2$ changing accordingly as shown by the DC-8 measurements during KORUS-AQ (Halliday et al., 2019) and indicated by model simulations (Tang et al. 2018). Tracking $CO_2$ and CO from fossil fuel emissions could be combined to further constrain fossil fuel emission fluxes.

Many studies have focused on the long-term CO emission trends now well characterized (Zheng et al., 2019). For the sake of forward modeling (see e.g. Huang et al., 2018), it is important to focus on improving the absolute emission totals and their spatio-temporal distribution. While bottom-up inventories are critical, the next step is a comparison of inverse modelling estimates in combination with aircraft observations (e.g. Gaubert et al., 2019) to assess transport, chemistry and deposition error. Multi-model estimates of the emissions will provide improved error bars on the CO budget, and hopefully reduced uncertainties from chemistry and meteorology.



## Appendix A: KORUS-AQ DC-8 instrumentation

CO and $CH_4$ were both measured using the fast-response (1 Hz), high-precision (0.1 % for $CH_4$, 1 % for CO) and high accuracy (2 %) NASA Langley Differential Absorption CO Measurement or DACOM (Sachse et al. 1987). Based on the differential absorption technique, CO and $CH_4$ were measured using an infrared tunable diode laser. The instrument has been used in many field campaigns and has been useful to evaluate profiles retrieved from satellite remote sensing of CO

(Warner et al., 2010; Tang et al., 2020). Formaldehyde was measured using the Compact Atmospheric Multispecies Spectrometer (CAMS), also at 1 Hz (Richter et al., 2015). NO, $NO_2$ and $O_3$ were measured by the NCAR chemiluminescence instrument (Ridley and Grahek 1990; Weinheimer et al., 1993). Nitric acid ($HNO_3$), hydrogen peroxide ($H_2O_2$) and methyl hydroperoxide ($CH_3OOH$) were measured using the California Institute of Technology Chemical

Ionization Mass Spectrometer (CIT-CIMS) (Crounse et al., 2006). Among the 82 speciated VOCs sampled by the discrete Whole Air Sampling (WAS) followed by multi-column gas chromatography (Simpson et al., 2020), we used ethyne ($C_2H_2$), ethane ($C_2H_6$), ethene ($C_2H_4$) and propane ($C_3H_8$). All the larger alkanes (i-butane, n-butane, i-pentane, n-pentane, n-hexane, n-heptane, n-octane, n-nonane, n-decane), alkenes (1-butene, i-butene, trans-3-butene and 1-3-

butadiene) and xylenes (mp-xylene, o-xylene) were summed (Table 3) for the comparison with the BIGALK, BIGENE and XYLENES respectively of the T1 surrogate species (Emmons et al., 2020). Methanol ($CH_3OH$), acetaldehyde ($CH_3CHO$), acetone ($CH_3COCH_3$), benzene ($C_6H_6$) and toluene ($C_7H_8$) were measured with the proton-transfer-reaction time-of-flight mass spectrometer (PTR-ToF-MS) at 10 Hz frequency (Müller et al., 2014). We also evaluate some meteorological

parameters, such as temperature and wind speed as well as water vapor moist volumetric mixing ratio measured by NASA open-path diode laser hygrometer (Podolske et al., 2003), with a 5% uncertainty. J values were measured using the CAFS instrument (Charged-coupled device Actinic Flux Spectroradiometer; Shetter and Müller, 1999; Petropavlovskikh et al., 2007).

## Appendix B: CAM-chem updates

### B1 $CH_4$ emissions from the Global Carbon Project $CH_4$

Radiatively active species, such as $CH_4$, are prescribed in CAM-chem using a latitudinal-monthly surface field derived from observations in the past and projections for the future, defined in the CMIP6 protocol (Meinshausen et al., 2017). In order to include the feedbacks in the $CH_4$-CO-OH

chemical mechanism, we choose to apply $CH_4$ emissions instead of the prescribed field. The scope of the paper is not to study the methane budget; the objectives are to see how much CO is produced from $CH_4$ during the campaign. The long-term goal is to get sensitivities to changes according to CO emission updates in order to analyze the feedbacks on $CH_4$ when CO is changed. We used emissions from some of the inversions of a recent compilation of $CH_4$ budget from top-down

estimates (Saunois et al., 2020). As a first step, we used the mean of the 11 inversions (Table B1) that assimilate $CH_4$ retrievals from the JAXA satellite Greenhouse Gases Observing SATellite (GOSAT, Kuze et al., 2009).

**Table B1: List of the 11 Methane inversions from the Global Methane Budget (Saunois et al., 2020), as indicated**
**by the number of inversions column. All the details are presented in the references.**





| Institution / Model | Observation Used | Number of inversions | References |
|---|---|---|---|
| FMI / CarbonTracker Europe-CH$_4$ | GOSAT NIES L2 v2.72 | 1 | Tsuruta et al. (2017) |
| LSCE & CEA / LMDz-PYVAR | GOSAT Leicester V7.2 | 2 | Yin et al. (2015) |
| LSCE & CEA / LMDz-PYVAR | GOSAT Leicester V7.2 | 4 | Yin et al., (2019) |
| NIES / NIES-TMFLEXPART (NTFVAR) | GOSAT NIES L2 v2.72 | 1 | Maksyutov et al. (2020); Wang et al. (2019) |
| TNO & VU / TM5-CAMS | GOSAT ESA/CCI v2.3.88 (combined with surface observations) | 1 | Segers (2020 report); Bergamaschi et al. (2010; 2013); Pandey et al., (2016) |
| EC-JRC / TM5-4Dvar | GOSAT OCPR v7.2 (combined with surface observations) | 2 | Bergamaschi et al., (2013, 2018) |

## B2 The HO$_2$ uptake by aerosol particles

The TS1 chemistry includes an HO$_2$ uptake by aerosol particles following the recommendation of Jaeglé et al. (2000) and Jacob et al. (2000), that form H$_2$O$_2$, with a reactive uptake coefficient $\gamma$ of 0.2, as follow:

$$HO_2 + aerosols \rightarrow 0.5 * H_2O_2 \qquad \text{with } \gamma = 0.2 \qquad \qquad (B1)$$

Based on Observations from the NASA Arctic Research of the Composition of the Troposphere from Aircraft and Satellites (ARCTAS) and other field campaigns, Mao et al. (2010, 2013) suggested a catalytic mechanism with transition metal ions (Cu and Fe) that rapidly converts HO$_2$ to H$_2$O instead of H$_2$O$_2$:

$$HO_2 + aerosols \rightarrow H_2O \qquad \text{with } \gamma = 0.2 \qquad \qquad (B2)$$

Using $\gamma = 1$ leads to a net loss of HO$_x$, which in turn increases CH$_4$ and CO lifetime and thus reduces the CO bias during the high latitude winter (Mao et al., 2013). Christian et al. (2017) simulated a range of possible values of $\gamma$ and evaluated the results against ARCTAS data and found that lower $\gamma$, closer to zero, gave a more realistic distribution of HO$_x$. Kanaya et al. (2009) studied ozone formation over Mount Tai, located in central East China, and looked at the possible influence of the heterogeneous loss of gaseous HO$_2$ radicals. They found that introducing the loss reduces HO$_2$ levels and increases ozone, with a more pronounced effect in the upper part of the boundary layer with less influence on the OH+NO$_2$+M reaction while the number density of aerosol particles is still important. Li et al. (2018) found this exact effect to explain the trends in the increase of ozone in Northern China, with the HO$_2$ uptake being the largest HOx sink in the upper boundary layer in this region. Thus, the initial comparison of CAM-chem using Eq. (B1) showed a large overestimation of H$_2$O$_2$. In a previous study using Eq. (B1), the increase in CO following data





assimilation increased hydrogen peroxide ($H_2O_2$) levels (Gaubert et al., 2016). Therefore, it is expected that the hydrogen peroxide ($H_2O_2$) would be severely overestimated if Eq. (B1) is used.

Miyazaki et al. (2019a) assimilated several satellite retrievals of chemical composition during KORUS-AQ, including MOPITT, and found a strong overestimation of $H_2O_2$ using Eq. (B1) in the chemical scheme of the MIROC-Chem model. Thus, the reaction in CAM-chem has been updated to Eq. (B2) with $\gamma=0.1$ prior to any data assimilation run.

**B3 Results on $HO_2$ uptake and methane emissions**

This section presents the results on the model update before the assimilation runs are conducted. Five CAM-Chem simulations were performed (Table B2), and CAM-Chem-Ref corresponds to the reference with prescribed $CH_4$ and Eq. (B2) for the $HO_2$ uptake. The CAM-H2O is performed with the update to Eq. 3 for the $HO_2$ uptake and the GCP-Ref is performed with the $CH_4$ emissions

instead of the $CH_4$ prescribed field. The GCP-H2O contains the update on $CH_4$ emissions and on the $HO_2$ uptake and has been run with $\gamma=0.2$ and $\gamma=0.1$.

**Table B2: description of the sensitivity test performed with CAM-Chem anterior to any assimilation run.**

| Simulation name | $HO_2$ uptake ($\gamma$) | Surface $CH_4$ |
|---|---|---|
| CAM-chem-Ref | Eq. (2) ($\gamma=0.2$) | Prescribed |
| CAM-Chem-H2O | Eq. (3) ($\gamma=0.2$) | Prescribed |
| GCP-Ref | Eq. (2) ($\gamma=0.2$) | Emissions |
| GCP-H2O ($\gamma=0.2$) | Eq. (3) ($\gamma=0.2$) | Emissions |
| GCP-H2O ($\gamma=0.1$) | Eq. (3) ($\gamma=0.1$) | Emissions |

Fig. B1 shows the average profiles for $H_2O_2$ and $CH_4$. There is a large bias in $H_2O_2$ for the reference simulation (CAM-chem-Ref) that is particularly large in the surface layer. The observed $H_2O_2$ at the surface is lower in the morning due to inhibited photochemical production and the nighttime deposition (Schroeder et al., 2020). Large model errors could then be due to uncertainties in the boundary layer height and wet deposition. However, this points to an underestimation of the

$H_2O_2$ dry deposition, a common feature found due to an overestimation of surface resistance (Ganzeveld et al., 2006; Nguyen et al., 2015). The $H_2O_2$ daytime deposition velocities calculated at the location of the Taehwa Research Forest site ranged between 0.4 cm.s$^{-1}$ and 1.3 cm.s$^{-1}$, which suggests an underestimation compared to the observed velocities of around 5 cm.s$^{-1}$ reported in the literature (Hall and Claiborn, 1997; Hall et al., 1999; Valverde-Canossa et al. 2006; Nguyen et al.,

2015). A simulation with a 5-fold increase of the $H_2O_2$ deposition velocity over land only partially reduces the $H_2O_2$ bias. Further work needs to be done to better understand the drivers of the $H_2O_2$ biases, which is beyond the scope of this study.

Interestingly, having $CH_4$ emissions (GCP-Ref) while keeping the original reaction (Eq. 2) gives

a slightly better $H_2O_2$, suggesting that using optimized emissions instead of a prescribed concentration field has an effect on the oxidants' distribution. The three simulations with the updated chemistry out-perform the references with biases almost halved. This is particularly true for the free troposphere. The modeled $H_2O_2$ profile seems rather insensitive to the choice of the $\gamma$ value. Since the simulations with the $\gamma=0.1$ performs slightly better, all following simulations will

be done with the updated reaction and $\gamma=0.1$. This is consistent with a recently published studies that diagnosed a median $\gamma$ value of 0.1 over the NCP region (Song et al., 2020).





Using emissions instead of fixed boundary conditions improves the simulated $CH_4$ near the surface, but with a lower tropospheric background (Figure A1). The comparison with $CH_4$ observations indicates a general underestimation. At this point, it is difficult to determine why it is underestimated.

A first reason could be a too strong $CH_4$ sink in the model compared to the sink considered in the inversions that derived the GCP emissions. However, the prescribed $CH_4$ is not resolved in longitude, while the difference for a given latitude can be up to 300ppb when using emissions (see Fig S1). Emissions also have uncertainties and could be underestimated, or may have just been estimated with lower OH than the one CAM-chem simulates for this period. Saunois et al. (2020) showed that the GOSAT based inversions have lower emissions than the surface-based inversions for the northern mid-latitudes. It is likely that the errors observed during KORUS-AQ are a 990 combination of both of those factors, as well as potential transport errors. Since the $CH_4$ profile is overall better reproduced with the GCP emissions, we have used the ensemble mean of the 11 GCP optimized emissions for the simulations presented in the main paper.

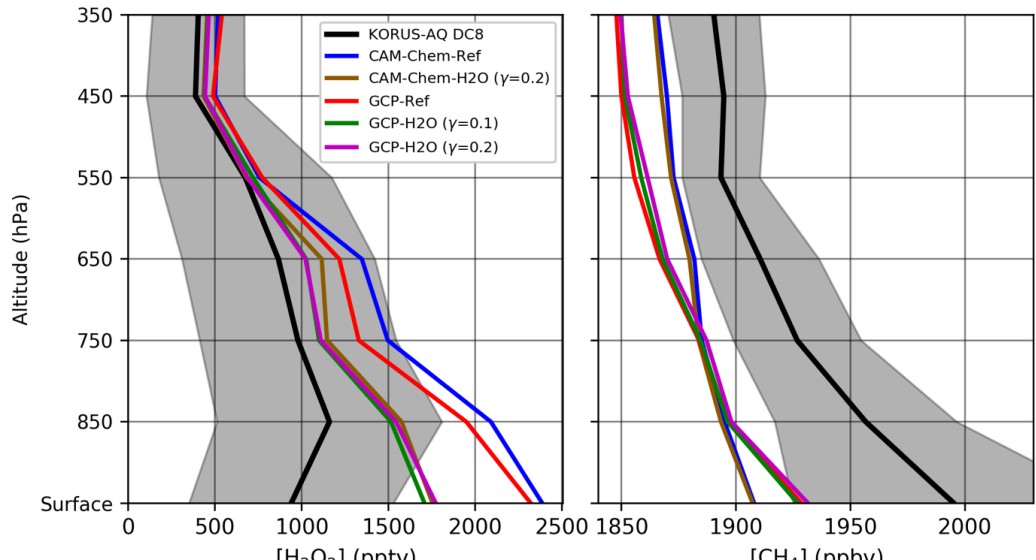

**Figure B1: Average $H_2O_2$ profiles (left panel) and $CH_4$ profiles (right panels) for all KORUS-AQ. The mean (black line) and standard deviation (shaded grey) of the DC-8 observations are calculated for each 100 hPa bins and only the mean is shown for model simulations.**

**Acknowledgments.** We would like to acknowledge high-performance computing support from Cheyenne (doi:10.5065/D6RX99HX) provided by NCAR's Computational and Information Systems Laboratory, sponsored by the National Science Foundation. Neither the European Commission nor ECMWF is responsible for any use that may be made of the information it contains. We thank Yi Yin, Arjo Segers, Aki Tsuruta, Peter Bergamaschi, Bo Zheng for sharing their $CH_4$ inversions. We acknowledge James H. Crawford, Glenn S. Diskin, Alan Fried, Andrew 1005 Weinheimer and everybody that contributed to the KORUS-AQ campaign. The PTR-MS



instrument team (P. Eichler, L. Kaser, T. Mikoviny, M. Müller, A. Wisthaler) is acknowledged for providing the PTR-MS data for this study. We also thanks Duseong Jo for reading the manuscript.

**Financial support.** This research has been supported by NASA (NNX16AD96G). This study was
also supported by NOAA's Climate Program Office's Modeling, Analysis, Predictions, and Projections program (NA18OAR4310283). The NCAR MOPITT project is supported by the National Aeronautics and Space Administration (NASA) Earth Observing System (EOS) Program. Observations made by Caltech were sponsored by NASA (NNX15AT97G). This material is based upon work supported by the National Center for Atmospheric Research, which is a major facility
sponsored by the National Science Foundation under cooperative agreement no. 1852977. The CESM project is supported primarily by the National Science Foundation (NSF). The ARIAs campaign was supported by the National Science Foundation (Grant # 1558259). Part of this work was conducted at the Jet Propulsion Laboratory, California Institute of Technology, under contract with the National Aeronautics and Space Administration (NASA). PTR-ToF-MS measurements
aboard the NASA DC-8 during KORUS-AQ were supported by the Austrian Federal Ministry for Transport, Innovation and Technology (bmvit) through the Austrian Space Applications Programme (ASAP) of the Austrian Research Promotion Agency (FFG).

**Code and datasets.** CESM2.1.0 is a publicly released version of the Community Earth System
Model and freely available online (at www.cesm.ucar.edu/, last access: 2 April 2020). The Data Assimilation Research Testbed is an open source software, code and documentation are available at https://dart.ucar.edu/ (DART, 2020). The Korea-United States Air Quality Field Study (KORUS-AQ) dataset is available at https://doi.org/10.5067/Suborbital/KORUSAQ/DATA01. The ARIAs observational dataset is available at https://www-air.larc.nasa.gov/missions/korus-
aq/index.html. MOPITT data are available at https://www2.acom.ucar.edu/mopitt. The Tropospheric Chemistry Reanalysis version 2 is available for download at https://tes.jpl.nasa.gov/chemical-reanalysis/products/monthly-mean/. The Copernicus Atmosphere Monitoring Service (CAMS) global bottom-up emission inventory is available on the Emissions of atmospheric Compounds and Compilation of Ancillary Data (ECCAD) web site
(https://eccad3.sedoo.fr).

**Competing interests.** The authors declare that they have no conflict of interest.

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
