# Peer review of "Correcting model biases of CO in East Asia: impact on oxidant distributions during KORUS-AQ"

_Atmospheric Chemistry and Physics, 2020_

## Referee Comment (RC1) · Anonymous Referee #1 · 12 Jul 2020

In this article, CO MOPITT data are assimilated in a global coupled chemistry-climate model (CAM-Chem) and the resulting distributions of CO and other compounds are evaluated against aircraft measurements from the KORUS and ARIAs campaigns conducted in May-June 2016 above South Korea and Northern China. In addition, the impact of different emission datasets on the model performance is evaluated with CAM-Chem using dynamical fields nudged to GEOS-FP analysis. Both the MOPITT assimilation and the comparisons with aircraft data suggest a substantial underestimation of CO emissions over Eastern China, consistent with several previous studies. Furthermore, the emission update improves (somewhat) the model performance for ozone and HOx, based on aircraft measurements (for O3) and box-model calculations (for HOx) constrained by measurements from the KORUS campaign. The paper is generally well-written, and the methodology is well described. A rather detailed discussion is provided, which covers not only the influence of emissions but also aspects related to dynamics, which I appreciated. The conclusions of the study appear mostly valid, although with some exaggeration of the benefits of assimilation and some oversimplication regarding specific aspects.

Major comments

- Most importantly, the role of model biases for NOx was clearly overlooked. The paper claims a good agreement against aircraft NOx data, whereas Fig. S2 shows a strong model underestimation (> factor of 2) of both NO and NO2 at all levels except the surface. This aspect warrants more discussion and possibly additional sensitivity calculations as it might have a strong impact on ozone, HOx and therefore CO and VOCs.

- l 106-115 "it has become evident (...) that several standard inventories of CO emissions in China are still too low (Tang et al 213; Yumimoto et al. 2014)" The two references cited are inappropriate for this statement. The first concerns only Beijing; and in the second study, the optimized CO emissions were found to be lower than their a priori inventory. Furthermore, Elguindi et al. (2020) showed that two widely used inventories (MEICv1.3 and REAS 2.1) have higher emissions than most top-down estimations. Therefore, I recommend to nuance the claim made repeatedly in the article that inverse modelilng suggest higher emissions than bottom-up inventories. Clearly, things are more complex as there is considerable variability among models (as well as among bottom-up inventories), and this should be acknowledged. I would be curious to know how the bottom-up emissions used in Cam-Chem compare with those intercompared by Elguindi et al. (2020). The top-down emissions determined in in this study using MOPITT CO depend on OH levels not only over S. Korea (for which we have partial constraints from KORUS) but also over East Asia and elsewhere. The related uncertainties deserve some discussion in this manuscript.

Minor comments

- the reference Feng et al. 2020 is cited (line 112) but is missing

- l 136-137 Note that De Smedt et al. (2015) (https://doi.org/10.5194/acp-15-12519-2015) reported negligible trends over Beijing and PRD over 2005-2016. Wang et al. (2015) (www.atmos-chem-phys.net/15/1489/2015) also reported decreased NMVOC traffic emissions over Beijing since 2002

l 142-145 "Ozone production and accumulation.." I don't see the link between this sentence and the rest of the paragraph

l 194-195 "This suggests an underestimation of emissions" Where does this come from?

l. 201-202 "two episodes of the transport phase": unclear

l. 230-231 "A high correlation between organic aerosol and CH2O, which is a characteristic of the importance of primary and secondary sources": very unclear, rephrase

l. 289 Where does this threshold of 20 ppbv comes from? How sensitive are the results to this value?

l. 317 Does the FINN inventory includes the emissions due to to agricultural residue burning, very high over the North China Plain in June ? Could an underestimation of those emissions in the model account for the low bias of CAM-Chem CO wrt MOPITT and aircraft ? It could be interesting to investigate whether the low bias is highest in June, compared to May and July. By the way, I don't think that the duration of the assimilation run is mentioned in the manuscript. What days or months of data are used in the assimilation? More generally the assimilation setup could be more detailed.

l. 335-336 "They found missing PFT data over Seoul and a large sensitivity in PFTs to changes in temperature": unclear, I don't understand what is meant here.

l. 341 "to determine a best fit to the observations...": the model does not match well

the observations for e.g. methanol, acetaldehyde, methyl hydroperoxide.. In fact, the high emissions lead to a worsening of the vertical gradient for some species. Please rephrase. The sensitivity test is not uninteresting but it fails to prove that biogenic emissions have a moderate impact on CO.

l. 381 I suppose that the assimilation of MOPITT influences not only emissions but also meteorology. If so, this should be mentioned and maybe shortly discussed.

l. 442 Does this implies that e.g. NOx anthropogenic emissions are updated as well, similar to CO?

l. 443 "strongly correlated": or completely correlated?

l. 479 "We compare our emission estimates": confusing. In this paragraph you sometimes refer to emissions and sometimes to the the model-calculated concentrations. Please re-structure this section to clearly state what is compared to what.

l. 485 Please define what is a "tag"

l. 489 The CAMS inventory is not mentioned in Table 2

l. 504 Define DACOM

l. 511 "Because of the reduction of the CO in the middle troposphere (...) the RMSE in MOPITT-DA is reduced...": I don't see that, please check.

l. 534-535 "if true errors in the VOCs are not correlated to CO, only noise will be introduced": sorry, unclear.

l. 545 replace "or" by "and/or"

l. 555 " bias in OH leads to correlated errors between CO and alkanes that can be mitigated by including these species in the state vector": I understand, but even if the VOCs would not be the state vector, their concentrations would be modified (improved) by the optimization.

l. 589 Only a small part of the HO2 underestimation is explained by the CO uncderestimation

l. 595 Note that the reduced HCHO formation is due to lower OH, entailing reduced VOC oxidation rate

- Note that the J(NO2) underestimation could play a role in the O3 underestimation.

- Does the model have biogenic CO emissions? Dry deposition of CO?

- l. 627-628: Note that this overestimation could be partly due to an overestimation of biogenic VOC emissions over Southern China and Southeast Asia (cf. Souri et al., ACPD 2020).

l. 638 "the spatial patterns of the prior emissions are important..": nevertheless, the posterior differences (Fig 5f) are larger than the prior differences (Fig 5c) suggesting that the prior emission differences are not crucial.

l. 645-657: This increase is not at all visible on Fig. 5g

l. 652 'China' –> 'Central China'

l. 657 I suppose those numbers are for Northern China, right?

l. 684 'smaller': by how much?

l. 685-687 The posterior CO is still 50 ppbv below MOPITT-DA. Why is that, what does this tell us? I think this deserves more attention.

l. 697 I recommend moderating the claim of better O3 match.

l. 'It means that CO acts..." Of course, this is expected. Delete "and is more consistent with the observations"

l. 699-700 "10 ppbv in the free troposphere": are you sure? I see this in the PBL, not the FT.

l. 710 Make a reference to Table 4 and mention that the average is for all levels.

l. 716-717 "such as errors in transport or chemistry": strange statement! How well does the assimilation matches MOPITT CO? If the match is very good (which I expect), then the discrepancy is due to a combination of measurement or representativity issues.

l. 738-740 "This suggests that weather patterns and direct anthropogenic emissions explain most of the CO variability during the campaign": I'm not so sure, since the posterior model run does not perform very well. The fact that the MOPITT assimilation run performs well indicate that the assimilation of initial conditions is important, which does not really tell mcuh about the reasons for CO variability. Is there a temporal evolution of CO emissions during the campaign period in the MOPITT-DA run?

l. 766 "for higher altitudes": at what altitudes?

l. 767 I would rather say "of stratospheric influence" or something like that.

l. 782-791: As mentioned above, the model discrepancies for NOx are very relevant for the discussion. I recommend to display the NOx comparisons for the separate phases. In Fig S2, the NOx are clearly much too low between 850 and 650 hPa, which might explain the HO2 biases. Note that the good agreement for OH might therefore be fortuitous, since it could be affected by higher NOx.

l. 803-804 "chemical production and loss via OH reaction from emissions..." unclear

l. 926 There is a net loss of HOx for all non-zero gamma values...

l. 932-933 "with a more pronounced effect in the upper part of the boundary layer with less influence on the OH+NO2+M reaction while..." unclear

l. 935 "this exact effect" also unclear

Technical comments/language

- l. 187 "looked"??

l. 247 Capitalize "we"

l. 277-278 "Their analysis of the aerosol pollution was mostly located below..." unclear, please rephrase

l. 309 Delete "Therefore, "

Equation 4 and following text: please use consistent notations/fonts for mathematic expressions

- l. 523 A parenthesis is missing

l. 633 remove final 's' in Plains

l. 642 & 643: '6' should be '5'

l. 645 Compare –> Compared

l. 653 The sentence is weird. The verb should be after the parenthesis. I suggest deleting 'is two times higher than'

l. 661 replace 'than for' by 'and'

l. 691 'above 900 hPa' is ambiguous, rephrase

l. 695 The layer?

l. 712 'means' –> "implies'

l. 763 "for" –> "to"

l. 767 'from' –> 'of'

- at various places in the manuscript and supplement, write "methanol" and not "Methanol", etc. for all species.

---

## Referee Comment (RC2) · Anonymous Referee #2 · 28 Jul 2020

This study investigates the impact of model biases in CO on the ability to simulate oxidants observed during the KORUS-AQ campaign. In particular, it explores the effects of assimilating MOPITT CO as well as using different emissions inventories. The question of how improving simulated CO alters other aspects of atmospheric chemistry is an interesting one, and the study applies state-of-the art tools to address it. The methodology is rigorous, but I have some suggestions, described in the general and specific comments, for how the results could be presented more clearly.

General Comments

The study includes quite a few model simulations, both with and without MOPITT assimilation and different prior or posterior anthropogenic and biogenic emissions inventories. Further simulations with different biogenic emissions are also discussed in the

supplemental material. While I understand the need for multiple sets of simulations to explore different ways of addressing CO bias, it can be difficult for the reader to keep track of what is included in each one and which lines to compare on figures (e.g. Fig. 9) showing many different model simulations. I suggest that the discussion in Section 6 (and some of the figures) would be easier to follow if that section were divided into two parts, with one part focused on the impact of MOPITT data assimilation, and the other part focused on the role of the emissions inventories.

The manuscript devotes as substantial amount of text to describing aspects of the model and assimilation system that do not seem directly tied to the discussion of the results. Some of this information, such as Section 4.2, could be condensed and/or moved to the Supplement or Appendix.

Appendix B contains a considerable amount of information and analysis, but I did not find it referenced in the main text.

The evaluation of NOx and other species in Section 2 of the supplement, and Fig. S2 in particular, is quite relevant to the interpretation of the results. It would be helpful to include some of this material in the main text.

Specific Comments

Line 62: Is "pollution ozone" the same as "ozone pollution"?

Line 460: Why is met data assimilation used in one set of runs and nudging in another set? Does this lead to transport differences between the two sets of runs?

Table 2: The simulation names should be made more informative/intuitive. For example, only one is called "Posterior" even though multiple runs use Posterior emissions.

Line 557: Does this mean it's the old RMSE minus the new one, or something else?

Lines 591-592: Please elaborate on this statement.

Line 598: I expect that biases in NOx, PAN, etc. can strongly impact the comparison

of the simulated oxidants, so I think this point deserves more discussion in the body of the text. Also, Fig. S2 shows that the MOPITT-DA makes H2O2 worse, and this should also be mentioned.

Line 659: Can you speculate on why the emissions are underestimated?

Line 698: Aren't VOCs also higher in this simulation? If so, how do you attribute the ozone production to CO rather than other VOCs?

Line 746: The definition of the tags needs to be explained somewhere.

Line 777: Does this mean there is a stratospheric intrusion reaching all the way to the surface? Or just more stratospheric/upper tropospheric influence somewhere in the profile?

Fig. 9: It would be helpful to include panels with the CO profile here so readers don't have to keep referring back to Fig. 8.

Editorial Comments

Line 204: "efficient" or "effective"?

Line 596: Sentence needs rewording

Lines 652-654: Sentence needs rewording.

Line 766: It would be clearer to avoid the parentheses and say lower OH and higher ozone.

---

## Author Comment (AC1) · 1 Oct 2020

Response to Anonymous Referee #1.

We thank the reviewer for the constructive comments and for providing suggestions to improve the manuscript. Our responses are shown in blue and added text is shown in italics.

In this article, CO MOPITT data are assimilated in a global coupled chemistry-climate model (CAM-Chem) and the resulting distributions of CO and other compounds are evaluated against aircraft measurements from the KORUS and ARIAs campaigns conducted in May-June 2016 above South Korea and Northern China. In addition, the impact of different emission datasets on the model performance is evaluated with CAMChem using dynamical fields nudged to GEOS-FP analysis. Both the MOPITT assimilation and the comparisons with aircraft data suggest a substantial underestimation of CO emissions over Eastern China, consistent with several previous studies. Furthermore, the emission update improves (somewhat) the model performance for ozone and HOx, based on aircraft measurements (for O3) and box-model calculations (for HOx) constrained by measurements from the KORUS campaign. The paper is generally well-written, and the methodology is well described. A rather detailed discussion is provided, which covers not only the influence of emissions but also aspects related to dynamics, which I appreciated. The conclusions of the study appear mostly valid, although with some exaggeration of the benefits of assimilation and some oversimplification regarding specific aspects.

Major comments
- Most importantly, the role of model biases for NOx was clearly overlooked. The paper claims a good agreement against aircraft NOx data, whereas Fig. S2 shows a strong model underestimation (> factor of 2) of both NO and NO2 at all levels except the surface. This aspect warrants more discussion and possibly additional sensitivity calculations as it might have a strong impact on ozone, HOx and therefore CO and VOCs.

Thanks, as suggested by reviewer 2 as well, we completely rewrote the section 5.2, and added references to Fig. S2 and Appendix B. the evaluation of NOx is now included in the figure 4 and is discussed in the text. We copied the paragraph here:

*"NO and $NO_2$ are reasonably well modeled for the surface layer, but are underestimated above, with a large underestimation at 850 hPa. The underestimation of $NO_x$ might explain the underestimation of $HO_2$. Additional comparison with $HNO_3$, $J(O_3)$, $J(NO_2)$ and $H_2O_2$ and PAN are shown in Figure S2. It suggests that the underestimation of $NO_x$ could be due to the underestimation of $J(NO_2)$ and the overestimation of $HNO_3$. Despite the update of the $HO_2$ heterogeneous uptake reaction and coefficient presented in appendix B, the CO increase leads to higher levels of $H_2O_2$, and the bias is therefore higher in the MOPITT-DA than the Control-Run (Figure S2). A lower value of the $HO_2$ heterogeneous uptake coefficient than the one used here ($\gamma=0.1$) might produce better results by reducing the $HO_2$ sink (see Appendix B). It suggests that errors in $NO_x$ and related chemistry drive the underestimation of $HO_2$ and of the sum of OH and $HO_2$ ($HO_x$). Overall, $HO_x$ is underestimated, and OH is fairly well simulated. This suggests that the CO chemical sink alone cannot explain the CO underestimation during the campaign. Alternatively, $CH_2O$ is underestimated in both simulations, suggesting an underprediction of the chemical production of secondary CO. A similar effect to that described in Gaubert et al. (2016) is shown, where an increase in CO through the sequential assimilation, leads to reduced OH and is slowing down of the VOC oxidation rate and formaldehyde formation, albeit a small effect. In the lower part of the atmosphere, the oxidation of additional CO leads to more effective ozone production and no changes above, consistent with observations. While the errors in $NO_x$ are important, the low $CH_2O$*

*points to a missing source, which could be due to an underestimation of CH₄ as well as NMVOCs (Appendix B).*

- l 106-115 "it has become evident (...) that several standard inventories of CO emissions in China are still too low (Tang et al 213; Yumimoto et al. 2014)" The two references cited are inappropriate for this statement. The first concerns only Beijing; and in the second study, the optimized CO emissions were found to be lower than their a priori inventory. Furthermore, Elguindi et al. (2020) showed that two widely used inventories (MEICv1.3 and REAS 2.1) have higher emissions than most top-down estimations.
Therefore, I recommend to nuance the claim made repeatedly in the article that inverse modelilng suggest higher emissions than bottom-up inventories. Clearly, things are more complex as there is considerable variability among models (as well as among bottom-up inventories), and this should be acknowledged. I would be curious to know how the bottom-up emissions used in Cam-Chem compare with those intercompared by Elguindi et al. (2020). The top-down emissions determined in in this study using MOPITT CO depend on OH levels not only over S. Korea (for which we have partial constraints from KORUS) but also over East Asia and elsewhere. The related uncertainties deserve some discussion in this manuscript.

Thank you for your comment, we added more nuances as you can read in the new version of the manuscript with discussion of the related uncertainties. We agree that the large uncertainties in inverse and forward modelling, in particular with regard to chemistry (OH sink, chemical oxidation of NMVOCs), preclude accurately disentangling the reasons for the CO underestimation in China. In fact, this is one of the main motivations of this study. However, we believe the manuscript is already long and includes comparison with other top-down and bottom-up estimates already. The comparison with other studies will be the purpose of future work with longer dedicated runs (for at least a year to cover the seasonal cycle).
The sentence has been corrected to: *Regionally, comparison with in-situ observations of forward and inverse modeling approaches suggests that several standard inventories of CO emissions in China are too low (e.g. Kong et al., 2020; Feng et al., 2020).*
Please note that the two studies we now discussed in this paragraph are using the MEIC inventory (with higher emissions as you pointed out) and still suggests that emissions are being underestimated overall, with regional and temporal exceptions.

Minor comments
- the reference Feng et al. 2020 is cited (line 112) but is missing
We added the reference.

- l 136-137 Note that De Smedt et al. (2015) (https://doi.org/10.5194/acp-15-12519-2015) reported negligible trends over Beijing and PRD over 2005-2016. Wang et al. (2015) (www.atmos-chem-phys.net/15/1489/2015) also reported decreased NMVOC traffic emissions over Beijing since 2002

We updated the text:
*As opposed to NOx emissions that have been decreasing since 2010, inventories suggest a net NMVOCs emissions increase (Zheng et al., 2018). While there are regional differences and no trends were observed in satellite retrievals of CH2O for the period 2004 to 2014 over Beijing and in the PRD (De Smedt et al., 2015), a more recent study suggests an overall increase of VOC emissions in the NCP by ~25 % between 2010 and 2016 (Souri et al., 2020). Shen et al., (2019) show that CH2O columns have a positive trend in urban regions of China from 2005 to 2016.*

l 142-145 "Ozone production and accumulation.." I don't see the link between this sentence and the rest of the paragraph

We rephrase the sentence to:

*"The transport of ozone pollution between source regions makes it difficult to correlate trends in ozone with the trends in emissions of its precursors (Wang et al., 2017)."*

l 194-195 "This suggests an underestimation of emissions" Where does this come from?
We agree this was unclear, we remove the sentence.

l. 201-202 "two episodes of the transport phase": unclear
We changed it to: *CO during two studied pollution events.*

l. 230-231 "A high correlation between organic aerosol and CH2O, which is a characteristic of the importance of primary and secondary sources": very unclear, rephrase
We removed the sentence.

l. 289 Where does this threshold of 20 ppbv comes from? How sensitive are the results to this value?

The 20 ppbv of SO2 corresponds to the 95th percentile of the whole campaign. It is arbitrarily designed to remove the high values of CO of around 500 ppbv and more (see Fig. 5 of Benish et al., 2020; https://www.atmos-chem-phys-discuss.net/acp-2020-194/). The inclusion of the values above the 95th percentiles are shifting the mean from 400 ppbv to 500 ppbv.
As you can see below, both observations mean and model mean have a higher CO without the filter. The overestimation seen for the inversions at the surface in the original plot is less important after removing the filter.

Figure 7 top panel, original:

[Figure]

[CO] (ppbv)     [O₃] (ppbv)

Figure 7 top panel, no threshold:

[Figure]

l. 317 Does the FINN inventory includes the emissions due to to agricultural residue burning, very high over the North China Plain in June? Could an underestimation of those emissions in the model account for the low bias of CAM-Chem CO wrt MOPITT and aircraft? It could be interesting to investigate whether the low bias is highest in June, compared to May and July.

By the way, I don't think that the duration of the assimilation run is mentioned in the manuscript. What days or months of data are used in the assimilation? More generally the assimilation setup could be more detailed.

*FINN includes agricultural residue burning, when it is detected by MODIS. Analysis of the aircraft data show little influence from biomass burning overall, although with some influence from Siberian fires (Russia) in June. While there were large fires in South East Asia in April, the plumes from them were over the Pacific and did not impact the Korean peninsula, (i.e. were not sampled during the campaign), see for instance Tang et al., 2019. We added the following:*

*The ensemble spin-up starts on April 1 2016 with perturbed emissions described above and with a spread in nudging parameters to perturb the dynamics. After a week, on April 7 2016, the Control-Run ensemble is initialized from the spin-up, this simulation is not nudged and this period is used to spin-up the inflation parameters for the assimilation of the weather observations only. The MOPITT-DA run is initialized from the Control-Run ensemble on April 15 2016.*

*The simulation ends on June 11 2016. It is not easy to compare the different months. Comparing different months and trying to attribute the sectoral sources will be the subject of future work.*

l. 335-336 "They found missing PFT data over Seoul and a large sensitivity in PFTs to changes in temperature": unclear, I don't understand what is meant here.

*We rephrase the sentence to:*

*"They found large sensitivities of calculated biogenic emissions to 3 different PFT datasets over Seoul, which resulted in local but significant changes in simulated O3."*

l. 341 "to determine a best fit to the observations...": the model does not match well the observations for e.g. methanol, acetaldehyde, methyl hydroperoxide.. In fact, the high emissions lead to a worsening of the vertical gradient for some species. Please rephrase. The sensitivity test is not uninteresting but it fails to prove that biogenic emissions have a moderate impact on CO.

Thanks, the truth is to determine the best fit to the observations of $CH_2O$, to check whether the CO bias is mainly of chemical origin, since most of secondary CO is from $CH_2O$ oxidation. The chosen simulation has the lowest $CH_2O$ bias at the surface.

We change the sentence to:

*We perform a set of simulations by varying biogenic emissions to determine the best fit to the observations of formaldehyde (CH2O) at the surface (see SI).*

l. 381 I suppose that the assimilation of MOPITT influences not only emissions but also meteorology. If so, this should be mentioned and maybe shortly discussed.

This point is addressed in the Section 4.5 (Variable localization and parameter estimation).

We understand the paragraph was not clearly worded (see also next comment), we corrected/rephrased the following:

*"This strict variable localization means that the assimilation of MOPITT only corrects the chemical state vector (i.e. CO) and has no impact on the meteorological state vector (U, V, T, Q, Ps) and vice-versa. However, we made an exception and extended our chemical state vector by including CO emissions from BB and anthropogenic sources separately and several NMVOCs. We added C2H2, C2H4, C2H6, C3H8, benzene, toluene, and the XYLENES, BIGENE and BIGALK surrogate species to the state vector. The NMVOCs with a strong anthropogenic and/or BB origin that have a primary sink with OH should be strongly correlated with CO (Miyazaki et al., 2012). The relationships between NMVOCs and CO leads to a correlation in their errors, so that the correlation existing in the ensemble will reflect those true errors."*

l. 442 Does this implies that e.g. NOx anthropogenic emissions are updated as well, similar to CO?

Only the emissions of CO are updated. For clarity we changed the acronym SF to SFCO:

*"In addition to the initial spread described above, spatially and temporally varying adaptive inflation is also applied to the optimized CO Surface Flux (SFCO) model variable during the analysis procedure."*

*"The relative increments obtained from the analysis in the form of the surface fluxes model variable (SFCO) is propagated back to the input files emissions (E) following:"*

And in the equation:

$$E_i^{analysis} = E_i^{prior}\left(1 + w\frac{\Delta SFCO_i}{SFCO_i}\right) \tag{2}$$

l. 443 "strongly correlated": or completely correlated?

The perturbations are completely correlated. We updated the sentence to:

*"This means the added noise in emissions of NMVOCs and CO from the BB or anthropogenic sectors will be completely correlated."*

l. 479 "We compare our emission estimates": confusing. In this paragraph you sometimes refer to emissions and sometimes to the model-calculated concentrations. Please re-structure this section to clearly state what is compared to what.

As suggested by the second reviewer as well, we divided the section 6 in two. The section 6 is now: **Comparison of anthropogenic emission estimates** and section 7 is: **Evaluation of the simulated vertical profiles against ARIAs and KORUS-AQ**.

In Sect. 6, we compare our bottom-up and top down estimate with other independent emission estimates (2 bottom-up and 1 top-down).

In section 7, we compare the concentration with CAM-Chem simulations using those various emission estimates (with prefix CAM_) that are described and labelled clearly in section 4.6 (see response to the next two comments).

l. 485 Please define what is a "tag"
We completely rewrote the paragraph.
That sentence you refer to now reads:
*"Note that the simulations denoted as CAM_HTAP (TCR-2 Prior) and CAM_TCR-2 (TCR-2 Posterior) are CAM-Chem simulations with the respective anthropogenic CO emissions from TCR-2."*
In the end of the paragraph we added:
*"We included artificial CO tracers or "CO tags", to track the anthropogenic contribution from different geographic area sources (e.g., Gaubert et al., 2016)."*

l. 489 The CAMS inventory is not mentioned in Table 2
We updated the table 2 with all the simulations:

**Table 2: Summary of the simulations. The Nudging (GEOS) refers to a CAM-Chem deterministic runs with specified dynamics, using a nudging to GEOS-FP analysis winds and temperatures (see supplement). Aside from the DART simulations (first 2 rows), all the simulations have the same initial conditions and the same nudging and only change by their anthropogenic CO emissions inputs.**

| Simulation name | Meteorology | Emissions (prior) |
|---|---|---|
| Control-Run | Assimilation (DART) | Prior (CEDS-KORUS-v5) |
| MOPITT-DA | Assimilation (DART) | Optimized (CEDS-KORUS-v5) |
| CAM_Kv5 | Nudging (GEOS) | Prior (CEDS-KORUS-v5) |
| CAM_HTAP | Nudging (GEOS) | Prior (HTAP v2) |
| CAM_MOP | Nudging (GEOS) | Posterior (CEDS-KORUS-v5) |
| CAM_MOP-Bio | Nudging (GEOS) | Posterior (CEDS-KORUS-v5) + MEGANx2 (see SI) |
| CAM_TCR-2 | Nudging (GEOS) | Posterior (TCR-2, HTAP v2) |
| CAM_CAMS | Nudging (GEOS) | CAMS (CAMS-GLOB-ANTv3.1) |

l. 504 Define DACOM
We rephrase to: *"We use the fully independent DC-8 Differential Absorption CO Measurement (DACOM)"*

l. 511 "Because of the reduction of the CO in the middle troposphere (...) the RMSE in MOPITT-DA is reduced...": I don't see that, please check.
We rephrase the sentence to: *"The RMSE in MOPITT-DA is reduced by around 10 ppbv compared to the Control-Run for the free troposphere (700 hPa to 300 hPa, Fig. 2)."*

l. 534-535 "if true errors in the VOCs are not correlated to CO, only noise will be introduced": sorry, unclear.
We rewrote the whole paragraph:

*"Concentrations of some VOCs have been added to the state vector and are therefore optimized, according to the covariance estimated by the ensemble, when MOPITT observations are assimilated. This setup will only provide meaningful corrections if CO and VOCs errors are highly correlated through common atmospheric and emission processes and if the ensemble samples those errors in the background error covariance. In this case VOCs analysis errors should be reduced by assimilating MOPITT CO, even though VOCs are not directly observed."*

l. 545 replace "or" by "and/or"
Done

l. 555 " bias in OH leads to correlated errors between CO and alkanes that can be mitigated by including these species in the state vector": I understand, but even if the VOCs would not be the state vector, their concentrations would be modified (improved) by the optimization.
We agree, this has been a part of our previous work where we look at some VOCs that were changing through forecast steps when CO is assimilated, without any state vector augmentation (Gaubert et al., 2016). Ethane and Propane concentration were increased because of the reduced OH.
We looked at the increments and there are differences at the analysis step.

l. 589 Only a small part of the HO2 underestimation is explained by the CO underestimation
We stated it carefully already: "This suggests that a small part of the HO2 underestimation can be explained by the CO underestimation."

l. 595 Note that the reduced HCHO formation is due to lower OH, entailing reduced VOC oxidation rate
We rephrase the sentence to:
*"A similar effect to that described in Gaubert et al. (2016) is shown, where an increase in CO through the sequential assimilation leads to reduced OH and is slowing down the VOC oxidation rate and formaldehyde formation, albeit a small effect."*

- Note that the J(NO2) underestimation could play a role in the O3 underestimation.
We completely rewrote the paragraph and added J(NO2) and HNO3 discussion to explain the large errors observed in NOx vertical profiles (as discussed above).

- Does the model have biogenic CO emissions? Dry deposition of CO?
The model has biogenic CO emissions calculated with MEGAN and dry deposition (see Lamarque et al., 2012) with no dry deposition over the forest Plant Functional Type in CLM (Müller and Brasseur, 1995).

- l. 627-628: Note that this overestimation could be partly due to an overestimation of biogenic VOC emissions over Southern China and Southeast Asia (cf. Souri et al., ACPD 2020).
We are not sure to follow the reviewer's point here since L627-628 does not mention biogenic VOCs, however we mentioned biogenic VOC below (regarding the L638 comment). Southeast Asian sources have not impacted Korea significantly during the campaign (e.g. Tang et al. 2019, JGR).

l. 638 "the spatial patterns of the prior emissions are important..": nevertheless, the posterior differences (Fig 5f) are larger than the prior differences (Fig 5c) suggesting that the prior emission differences are not crucial.
We agree, the sentence now reads: "Prior emissions of CO, biogenic and anthropogenic VOCs and NOx can all contribute to differences between the TCR-2 and our DART/CAM-chem estimate."

l. 645-657: This increase is not at all visible on Fig. 5g
Since it is hard to tell on the figure, we calculated the exact change the SMA pixel (34 %) and updated the text to:
*Compared to its prior, the DART/CAM-Chem posterior emissions are increased by 25 % for South Korea, and by 34 % over the SMA.*

l. 652 'China' –> 'Central China'
Done

l. 657 I suppose those numbers are for Northern China, right?
It is 33 % for Central China and 80 % for Northern China, the sentence now reads:
On average, the increase in emissions due to assimilation is about 33 % for Central China and nearly doubled (80 %) in Northern China, from 2.7 $TgCO.Month^{-1}$ to 4.9 $TgCO.Month^{-1}$.

l. 684 'smaller': by how much?
For the surface layer, it is smaller by 60 % for TCR-2 and by 30 % for MOPITT-DA (as you can see from the figures above). We updated the text:
*"The MOPITT-DA and the TCR-2 overestimate the CO concentrations compared to the measurements for this surface layer although this overestimate is smaller by 60 % for TCR-2 and by 30 % for MOPITT-DA when a value higher than 20 ppbv SO2 (the approximate 95th percentile) is used to define plumes for exclusion."*

l. 685-687 The posterior CO is still 50 ppbv below MOPITT-DA. Why is that, what does this tell us? I think this deserves more attention.
We added: *"While both simulations do not have exactly the same transport, the remaining underestimation is likely to be due to the sequential data assimilation in the MOPITT-DA runs that compensate for the remaining biases."*

l. 697 I recommend moderating the claim of better O3 match.
We rewrote the sentence to:
*For this layer, higher O3 was found for simulations with higher CO. While it suggests that reducing CO biases can improve O3, NO2 and NMVOCs such as aromatics seem to play an important role in the ozone formation in the region (Benish et al., 2020). The mean O3 concentration is still underestimated by around 10 ppbv in the free troposphere.*

l. 'It means that CO acts..." Of course, this is expected. Delete "and is more consistent with the observations"
Done

l. 699-700 "10 ppbv in the free troposphere": are you sure? I see this in the PBL, not the FT.
We change the sentence to: "10 ppbv for the other vertical layers", since it applies both above and below in altitude.

l. 710 Make a reference to Table 4 and mention that the average is for all levels.
Done

l. 716-717 "such as errors in transport or chemistry": strange statement! How well does the assimilation matches MOPITT CO? If the match is very good (which I expect), then the discrepancy is due to a combination of measurement or representativity issues.

We agree it was not clear. The paragraph now reads:

*Correcting only the bias in anthropogenic emissions is not as efficient as the joint optimization of anthropogenic emissions and sequential optimization of initial conditions through data assimilation (MOPITT-DA). It suggests that other sources of errors such as transport and chemistry can be mitigated by state assimilation. The MOPITT-DA has an average CO of 179 ppbv, resulting in 12 % underestimation on average (Table 4), which is well between the range in measurement and representativeness errors.*

l. 738-740 "This suggests that weather patterns and direct anthropogenic emissions explain most of the CO variability during the campaign": I'm not so sure, since the posterior model run does not perform very well. The fact that the MOPITT assimilation run performs well indicate that the assimilation of initial conditions is important, which does not really tell mcuh about the reasons for CO variability. Is there a temporal evolution of CO emissions during the campaign period in the MOPITT-DA run?

This is true. We corrected the sentence: "*Updating the anthropogenic emissions from the bottom-up to the top-down inventories improved the representation of the CO anomalies. This suggests that weather patterns and the direct anthropogenic emissions explain some of the CO variability during the campaign. However, since the MOPITT-DA simulation is reproducing the anomalies, it suggests that chemistry and transport are important too.*"

This DA system cannot resolve the monthly temporal variations accurately because of the chosen time window (sigma ~ 2months) for the inversion.

l. 766 "for higher altitudes": at what altitudes?

We corrected the sentence to:

*This anomaly is reflected through the OH (and O3) vertical profiles that also follow respectively lower (higher) concentrations between 800 hPa and 400 hPa (Figure 9).*

l. 767 I would rather say "of stratospheric influence" or something like that.

We changed the sentence to:

"*This indicates rather clean air masses, probably with larger stratospheric contribution.*"

l. 782-791: As mentioned above, the model discrepancies for NOx are very relevant for the discussion. I recommend to display the NOx comparisons for the separate phases. In Fig S2, the NOx are clearly much too low between 850 and 650 hPa, which might explain the HO2 biases. Note that the good agreement for OH might therefore be fortuitous, since it could be affected by higher NOx.

The sentence has been rewritten to:

*The OH is overestimated because of a lack of CO, other VOCs and/or errors in the vertical profile of NOx.*

l. 803-804 "chemical production and loss via OH reaction from emissions..." unclear

We rephrased to:

"*We also highlight in this work that errors in anthropogenic and biogenic VOCs, chemical production and loss and transport errors confound the attribution of this bias in current model simulations.*"

l. 926 There is a net loss of HOx for all non-zero gamma values...

We were referring to the change from Eq. B1 to Eq. B2, since H2O2 production is not an absolute net loss of HOx. We changed it to: "*Using Eq. B2 and $\gamma$=1 leads to a large loss of HOx, which in turn increases the CH4 and CO lifetime and thus reduces the CO bias during the high latitude winter (Mao et al., 2013).*"

l. 932-933 "with a more pronounced effect in the upper part of the boundary layer with less influence on the OH+NO2+M reaction while..." unclear

We rephrased to: *"They found that introducing the loss reduces HO$_2$ levels and increases ozone, with a more pronounced effect in the upper part of the boundary layer where the role of OH+NO2 +M reaction does not play a significant role in the radical termination reaction while the number density of aerosol particles is still important."*

l. 935 "this exact effect" also unclear

We changed the sentences to:

*"Li et al. (2018) found that the HO$_2$ uptake was the largest HOx sink in the upper boundary layer in northern China. They suggested that the reduction in HO$_2$ uptake caused by the decrease of aerosols was responsible for the increase of O$_3$ in the region."*

Technical comments/language

- l. 187 "looked"??

Replaced by "studied"

l. 247 Capitalize "we"

Done

l. 277-278 "Their analysis of the aerosol pollution was mostly located below..." unclear, please rephrase

Changed to "The aerosol pollution was mostly located ..."

l. 309 Delete "Therefore, "

Done

Equation 4 and following text: please use consistent notations/fonts for mathematic Expressions

Done

- l. 523 A parenthesis is missing

Done

l. 633 remove final 's' in Plains

We replaced North China Plain by NCP after the first occurrence.

l. 642 & 643: '6' should be '5'

Done

l. 645 Compare –> Compared

Done

l. 653 The sentence is weird. The verb should be after the parenthesis. I suggest deleting 'is two times higher than'

The sentence now reads: *The difference between CAMS (3.65 TgCO.Month-1) and the CEDS-KORUSv5 (5.7 TgCO.Month-1) is twice as high as the difference between DART/CAM-chem posterior (7.6 TgCO.Month-1) and TCR-2 (8.7 TgCO.Month-1).*

l. 661 replace 'than for' by 'and'

Done

l. 691 'above 900 hPa' is ambiguous, rephrase
We changed it to: *"For altitudes ranging between 900 hPa and 600 hPa,"*

l. 695 The layer?
Corrected to *"The 875 hPa (900 to 850 hPa) layer mean (and median)"*

l. 712 'means' –> "implies'
Done

l. 763 "for" –> "to"
Done

l. 767 'from' –> 'of'
Done

- at various places in the manuscript and supplement, write "methanol" and not "Methanol", etc. for all species.
Done

References:

Müller, J-F, and G. Brasseur, IMAGES, A three-dimensional chemical transport model of the global troposphere, J. Geophys. Res., https://doi.org/10.1029/94JD03254, 100, 16,445-16,490, 1995.

Tang, W., Emmons, L. K., Arellano Jr, A. F., Gaubert, B., Knote, C., Tilmes, S., et al.: Source contributions to carbon monoxide concentrations during KORUS-AQ based on CAM-chem model applications. Journal of Geophysical Research: Atmospheres, 124, 2796–2822. https://doi.org/10.1029/2018JD029151, 2019.

---

## Author Comment (AC2) · 1 Oct 2020

Response to Anonymous Referee #2.
We thank the reviewer for the constructive comments and for providing suggestions to improve the manuscript. Our responses are shown in blue and added text is shown in italics.

This study investigates the impact of model biases in CO on the ability to simulate oxidants observed during the KORUS-AQ campaign. In particular, it explores the effects of assimilating MOPITT CO as well as using different emissions inventories. The question of how improving simulated CO alters other aspects of atmospheric chemistry is an interesting one, and the study applies state-of-the art tools to address it. The methodology is rigorous, but I have some suggestions, described in the general and specific comments, for how the results could be presented more clearly.

General Comments
The study includes quite a few model simulations, both with and without MOPITT assimilation and different prior or posterior anthropogenic and biogenic emissions inventories. Further simulations with different biogenic emissions are also discussed in the supplemental material. While I understand the need for multiple sets of simulations to explore different ways of addressing CO bias, it can be difficult for the reader to keep track of what is included in each one and which lines to compare on figures (e.g. Fig. 9) showing many different model simulations. I suggest that the discussion in Section 6 (and some of the figures) would be easier to follow if that section were divided into two parts, with one part focused on the impact of MOPITT data assimilation, and the other part focused on the role of the emissions inventories.

The manuscript devotes as substantial amount of text to describing aspects of the model and assimilation system that do not seem directly tied to the discussion of the results. Some of this information, such as Section 4.2, could be condensed and/or moved to the Supplement or Appendix.

Appendix B contains a considerable amount of information and analysis, but I did not find it referenced in the main text.

The evaluation of NOx and other species in Section 2 of the supplement, and Fig. S2 in particular, is quite relevant to the interpretation of the results. It would be helpful to include some of this material in the main text.

Thanks, as suggested, we divided the section 6 in two. The section 6 is now: *Comparison of anthropogenic emission estimates* and section 7 is: *Evaluation of the simulated vertical profiles against ARIAs and KORUS-AQ*.
We completely rewrote the section 5.2, and added references to Fig. S2 and Appendix B. the evaluation of NOx is now included in Figure 4 and is discussed in the text.
There are new sentences that refer to appendix B, In section 3.1:
*"A summary of the model references is presented in Table 1. We have made some additional changes for this study, presented in Appendix B. In particular, we updated the heterogeneous uptake coefficient of HO2 and its coefficient."*

In Sect. 5.2, we added:

*"Despite the update of the HO2 heterogeneous uptake reaction and coefficient presented in appendix B, [...]"*

*"A lower value of the HO2 heterogeneous uptake coefficient than the one used here (γ=0.1) might produce better results by reducing the HO2 sink (see Appendix B)."*

*"While the errors in NOx are important, the low CH2O points to a missing source, which could be due to an underestimation of CH4 as well as NMVOCs (Appendix B)."*

Specific Comments

Line 62: Is "pollution ozone" the same as "ozone pollution"?
We changed it to ozone pollution.

Line 460: Why is met data assimilation used in one set of runs and nudging in another set? Does this lead to transport differences between the two sets of runs?
The assimilation is done using an Ensemble Adjustment Kalman Filter, that requires to run an ensemble of simulations, thus the assimilation is much more computationally expensive to perform.
The two assimilation runs perform well as described in Fig. S3, S4, S5, and overall provide a better meteorology than the nudging runs. Here is an update of table S1, now Table S2 with the Control-run added:

**Table S2: Overall statistics for Wind Speed, H2O, and Temperature.**

| Wind Speed | simulations | RMSE | r | Bias |
|---|---|---|---|---|
| | Control-Run | 8.44 | 0.67 | 1.02 |
| | prior-d-0.24 | 9.21 | 0.61 | 1.32 |
| | prior-d-0.48 | 8.96 | 0.63 | 1.22 |
| | prior-d-0.72 | 8.87 | 0.63 | 1.11 |
| | prior-m-0.24 | 8.89 | 0.64 | 1.32 |
| | prior-m-0.48 | 8.61 | 0.65 | 1.01 |
| | prior-m-0.72 | 8.56 | 0.66 | 0.92 |
| $H_2O$ | simulations | RMSE | r | Bias |
| | Control-Run | 2124.7 | 0.91 | -123.4 |
| | prior-d-0.24 | 2990.14 | 0.86 | 766.62 |
| | prior-d-0.48 | 2570.17 | 0.89 | 498.83 |
| | prior-d-0.72 | 2459.9 | 0.89 | 396.89 |
| | prior-m-0.24 | 2429.31 | 0.89 | 32.67 |
| | prior-m-0.48 | 2191.41 | 0.91 | -68.4 |
| | prior-m-0.72 | 2126.97 | 0.91 | -98.73 |
| Temperature | simulations | RMSE | r | Bias |
| | Control-Run | 2.65 | 0.99 | 1.09 |
| | prior-d-0.24 | 2.95 | 0.98 | 1.09 |

| | | | |
|---|---|---|---|
| **prior-d-0.48** | **2.88** | **0.98** | **1.03** |
| **prior-d-0.72** | **2.85** | **0.99** | **1.01** |
| **prior-m-0.24** | **2.88** | **0.98** | **0.96** |
| **prior-m-0.48** | **2.81** | **0.99** | **0.93** |
| **prior-m-0.72** | **2.79** | **0.99** | **0.94** |

We believe an entire study could be dedicated to this interesting topic of research.

Table 2: The simulation names should be made more informative/intuitive. For example, only one is called "Posterior" even though multiple runs use Posterior emissions.
We updated the Table 2 and simulation names as follows. We also made it clear in the text, and contrast those in section 6 which compare the emissions and section 7 which compares a suite of deterministic runs forced by different anthropogenic CO inventories.

*Table 2: Summary of the simulations. The Nudging (GEOS) refers to a CAM-Chem deterministic run with specified dynamics, using a nudging to GEOS-FP analysis winds and temperatures (see supplement). Aside from the DART simulations (first 2 rows), all the simulations have the same initial conditions and the same nudging and only change by their anthropogenic CO emissions inputs.*

| *Simulation name* | *Meteorology* | *Emissions (prior)* |
|---|---|---|
| *Control-Run* | *Assimilation (DART)* | *Prior (CEDS-KORUS-v5)* |
| *MOPITT-DA* | *Assimilation (DART)* | *Optimized (CEDS-KORUS-v5)* |
| *CAM_Kv5* | *Nudging (GEOS)* | *Prior (CEDS-KORUS-v5)* |
| *CAM_HTAP* | *Nudging (GEOS)* | *Prior (HTAP v2)* |
| *CAM_MOP* | *Nudging (GEOS)* | *Posterior (CEDS-KORUS-v5)* |
| *CAM_MOP-Bio* | *Nudging (GEOS)* | *Posterior (CEDS-KORUS-v5) + MEGANx2 (see SI)* |
| *CAM_TCR-2* | *Nudging (GEOS)* | *Posterior (TCR-2, HTAP v2)* |
| *CAM_CAMS* | *Nudging (GEOS)* | *CAMS (CAMS-GLOB-ANTv3.1)* |

Line 557: Does this mean it's the old RMSE minus the new one, or something else?
Yes, it does, but relative to the old RMSE. We updated the text in the figure captions:
*"The relative error change is the opposite of the difference in Root Mean Square Error relative to the Control-Run (i.e., (Control-Run-MOPITT-DA)/Control-Run). Thus, a positive relative error change means an improvement compared to the Control-Run."*

Lines 591-592: Please elaborate on this statement.
The paragraph has been updated and the sentence now reads:
*"It suggests that errors in NOx and related chemistry drive the underestimation of HO2 and of the sum of OH and HO2 (HOx). Overall, HOx is underestimated, and OH is fairly well simulated. This suggests that the CO chemical sink alone cannot explain the CO underestimation during the campaign."*

Line 598: I expect that biases in NOx, PAN, etc. can strongly impact the comparison of the simulated oxidants, so I think this point deserves more discussion in the body of the text. Also, Fig. S2 shows that the MOPITT-DA makes H2O2 worse, and this should also be mentioned.

We completely rewrote the section, added NOx comparison to the figure 4 and mentioned the figure S2 with and discussed the H2O2 issues. See also the responses to Reviewer 1.

Line 659: Can you speculate on why the emissions are underestimated?
The end of the paragraph now reads:
*"More work should be dedicated to check whether the assumptions made on the prior estimates impact the retrieved emissions. This includes improving the regional distribution and scaling up the baseline prior CO emissions, but also how much the model uncertainties in the OH chemical sink impact the CO inversions (e.g., Müller et al., 2018). A comparison of the amount of residential coal burning emissions in bottom-up inventories could help in understanding the discrepancy and quantifying potential offsets (Chen et al., 2017; Cheng M., et al., 2017; Zhi et al., 2017, Benish et al., 2020)."*

Line 698: Aren't VOCs also higher in this simulation? If so, how do you attribute the ozone production to CO rather than other VOCs?
VOCs are changed in the MOPITT-DA and in the CAM_MOP-Bio (aka CAM-Chem-Bio). The 5 simulations CAM_CAMS, CAM_Kv5 (DART-CAM-Chem prior), CAM_MOP (DART-CAM-Chem Posterior), CAM_HTAP (Prior TCR-2) and CAM_TCR-2 (posterior TCR-2) only differ by their CO anthropogenic emissions and share the same meteorology. The O3 is further improved with additional biogenic VOCs that fuel ozone production, as discussed in the following section.

Line 746: The definition of the tags needs to be explained somewhere.
In the end of section 4.6, we added:
*We included artificial CO tracers or "CO tags", to track the anthropogenic contribution from different geographic area sources (e.g. Gaubert et al., 2016).*

Line 777: Does this mean there is a stratospheric intrusion reaching all the way to the surface? Or just more stratospheric/upper tropospheric influence somewhere in the profile?
We believe you refer to L767, Weather phase 2 of the campaign. The CAM-chem stratospheric ozone tracer suggests enhancements of stratospheric O3 in the free troposphere, disconnected from the enhancement observed at the surface / in the boundary layer.

Fig. 9: It would be helpful to include panels with the CO profile here so readers don't have to keep referring back to Fig. 8.
We believe that the Fig. 8 should be placed close enough to Fig. 9 in the manuscript.

Editorial Comments
Line 204: "efficient" or "effective"?

Thanks, we changed efficient to effective.

Line 596: Sentence needs rewording
The new sentence now reads:
*"In the lower part of the atmosphere, the oxidation of additional CO leads to more effective ozone production and no changes above, consistent with observations."*

Lines 652-654: Sentence needs rewording.
The new sentences now read:
*"Our inversion suggests an underestimation of bottom-up emission inventories for China. The agreement between the posterior emissions for Central China is better than for the bottom-up inventory (Fig. 6)."*

Line 766: It would be clearer to avoid the parentheses and say lower OH and higher ozone.
The new sentence reads:
*"This anomaly is reflected through lower OH and higher $O_3$ between 800 hPa and 400 hPa (Figure 9)."*

---

## Author Comment (AC3) · 5 Oct 2020

Tilmes et al., (2020) was missing, the correct reference is Tilmes et al. (2019). Crutzen et al. 1979 was missing in the text. We added it where it was supposed to be added.

References Crutzen, P. J., L. E. Heidt, J. P. Krasnec, W. H. Pollack, and Seiler W.: Biomass burning as a source of atmospheric gases CO, H2, N2O, NO, CH3Cl, and COS, Nature, 282, 253-256, https://doi.org/10.1038/282253a0, 1979.

Tilmes, S., Hodzic, A., Emmons, L. K., Mills, M. J., Gettelman, A., Kinnison, D. E., Park, M., Lamarque, J.-F., Vitt, F., Shrivastava, M., Campuzano-Jost, P., Jimenez, J. L., and Liu, X.: Climate forcing and trends of organic aerosols in the Community Earth System Model (CESM2), J. Adv. Model. Earth Syst., 11, 4323–4351,

https://doi.org/10.1029/2019MS001827, 2019.

We added 2 new co-authors, they made the CO measurements during the KORUS-AQ campaigns.
* * *